# A neomorphic variant in SP7 alters sequence specificity and causes a high-turnover bone disorder

Julian C. Lui [1✉], Adalbert Raimann [2,3], Hironori Hojo[4], Lijin Dong[5], Paul Roschger[6], Bijal Kikani[1], Uwe Wintergerst [7], Nadja Fratzl-Zelman[3,6], Youn Hee Jee [1], Gabriele Haeusler [2,3] & Jeffrey Baron[1]

SP7/Osterix is a transcription factor critical for osteoblast maturation and bone formation. Homozygous loss-of-function mutations in *SP7* cause osteogenesis imperfecta type XII, but neomorphic (gain-of-new-function) mutations of *SP7* have not been reported in humans. Here we describe a de novo dominant neomorphic missense variant (c.926 C > G:p.S309W) in *SP7* in a patient with craniosynostosis, cranial hyperostosis, and long bone fragility. Histomorphometry shows increased osteoblasts but decreased bone mineralization. Mice with the corresponding variant also show a complex skeletal phenotype distinct from that of *Sp7*-null mice. The mutation alters the binding specificity of SP7 from AT-rich motifs to a GC-consensus sequence (typical of other SP family members) and produces an aberrant gene expression profile, including increased expression of *Col1a1* and endogenous *Sp7*, but decreased expression of genes involved in matrix mineralization. Our study identifies a pathogenic mechanism in which a mutation in a transcription factor shifts DNA binding specificity and provides important in vivo evidence that the affinity of SP7 for AT-rich motifs, unique among SP proteins, is critical for normal osteoblast differentiation.

[1] Section on Growth and Development, Eunice Kennedy Shriver National Institute of Child Health and Human Development, National Institutes of Health, Bethesda, MD, USA. [2] Department of Pediatrics and Adolescent Medicine, Division of Pediatric Pulmonology, Allergology and Endocrinology, Medical University of Vienna, Vienna, Austria. [3] Vienna Bone and Growth Center, Vienna, Austria. [4] Center for Disease and Integrative Medicine, University of Tokyo, Tokyo, Japan. [5] Genetic Engineering Core, National Eye Institute, National Institute of Health, Bethesda, MD, USA. [6] Ludwig Boltzmann Institute of Osteology at the Hanusch Hospital of OEGK and AUVA Trauma Centre Meidling, 1st Medical Department, Hanusch Hospital, Vienna, Austria. [7] Department of Pediatrics, Hospital of Braunau, Braunau, Austria. ✉email: luichunk@mail.nih.gov

SP7 is a transcription factor which is required for normal bone formation[1]. While RUNX2, a runt family transcription factor, is required for differentiation of mesenchymal stem cells into pre-osteoblasts[2,3], SP7 is required for the subsequent maturation of pre-osteoblasts into osteoblasts. Interestingly, genome-wide association studies (GWAS) have identified common genetic variants in *SP7* to be associated with bone mineral density[4–6]. Pathogenic variants in *SP7* have been described in two previous families[7,8] with recessive osteogenesis imperfecta (OI type XII), which is characterized by generalized osteoporosis. Similarly, homozygous knockout of *Sp7* in mice impairs osteoblast differentiation and bone formation.

Despite the physiological and clinical importance of SP7, much has yet to be learnt about its molecular function. Based on similarities with other SP family members, SP7 was first thought to bind GC-rich sequences[1,9–11] which are present in the promoters of many genes important for osteoblast functions[12,13], such as *Col1a1* and *Col1a2*[14]. However, other studies have shown evidence for a lack of binding preference for a GC-rich sequence[15]. More recently, Hojo et al. showed that SP7 differs from other SP proteins in that it has a lower binding affinity for GC-rich sequences, and instead interacts with DLX5 and with AT-rich promoter/enhancer sequences to transactivate genes in osteoblasts[16]. These findings were based on in vitro ChIP-Seq data using osteoblasts and thus it was not known whether the unique preference of SP7 to bind AT-rich motifs is physiologically important in vivo.

In this work, we studied a patient with a complex skeletal disease involving high bone turnover (increased bone formation and bone resorption) resulting in some regions of increased bone density and other regions of decreased bone density. We demonstrate that this disorder results from a *de novo*, neomorphic variant in SP7 which redirects the binding specificity of this transcription factor from AT-rich motifs to a GC-consensus sequence, resulting in aberrant gene expression and, consequently, abnormal osteoblast differentiation. We found that, in mice, the orthologous missense variant in *Sp7* produces a complex skeletal phenotype, confirming that the variant is pathogenic. Mice with this variant are dissimilar to *Sp7* knockout mice, both in inheritance and phenotype, confirming that the variant does not cause a simple loss of function. Our study thus identifies a novel pathogenic mechanism in which a variant in a transcription factor redirects DNA binding specificity. The findings also provide the first in vivo evidence that the preference of SP7 for AT-rich motifs, unique among SP proteins, is physiologically important for normal osteoblast differentiation.

## Results

***De novo* SP7 variant in a patient with complex bone disorder.** We evaluated a child with a complex skeletal disorder which included severe scoliosis, thickened calvarium, craniosynostosis, osteosclerosis of the clavicles and spine, and recurring fractures in the lower extremities (more than 15 fractures before age 6 years) (Fig. 1a–e). Long bones showed areas of both increased and decreased cortical thickness (Fig. 1f). The height-adjusted lumbar bone mineral density by DXA scan was distinctly increased (+5.8 SDS). He had normal serum calcium, phosphate, vitamin D, and parathyroid hormone, but elevated alkaline phosphatase (1793 IU/l, normal range 151–342 IU/l) and TRAP5b (>15.5 U/l, above detection range), suggesting both increased bone formation and resorption. Consistently, bone histomorphometry showed a marked increase in all indices of bone formation (osteoblast bone surface per total bone surface, osteoblast and osteoid surface per bone surface, and osteoid thickness) and of bone resorption (osteoclast surface per bone surface and eroded surface per bone surface) in trabecular bone of the affected patient compared to reference values (Fig. 1g–k, Supplementary Figs. 1, 2). These results are typical for elevated bone turnover. High bone turnover often results in decreased bone matrix mineralization because newly formed bone packets are removed before they archive full mineralization[17,18]. Consistent with this expectation, quantitative backscattered electron imaging (qBEI) of trabecular bone showed markedly decreased matrix mineralization with increased heterogeneity (Fig. 1l–n). Cortical bone showed a similar but less pronounced decrease in bone matrix mineralization. Polarized light microscopy revealed abnormal lamellar collagen fibril arrangement (Supplementary Fig. 3). The patient had normal dentition, intelligence, hearing, and stature. Collectively, the clinical presentation of our patient is distinct from previous known genetic disorders of high bone turnover, osteosclerosis or craniosynostosis. Urine deoxypyridinoline was in the age-specific normal range in two samples (10.2, 12.7 nmol/mmol creatinine, normal range 4.3–24.7) and urine hydroxyproline was below the normal range (0.0, 0.1 nmol/mmol creatinine, normal range 0–13.0). However, both of these markers of bone resorption and also the DXA measurements may have been affected by the long-term treatment with bisphosphonates which diminishes bone resorption.

Comparative genomic hybridization microarray did not detect any deletions or duplications. Exome sequencing identified in the proband a rare *de novo* missense variant in *SP7* (chr12:g.32723 30G>C (hg19), c.926C>G:p.S309W, NM_001173467.2), which was confirmed by Sanger sequencing (Supplementary Fig. 4). This genetic variant was not reported in gnomAD and was predicted to be deleterious to protein function by multiple in silico analyses (CADD, SIFT, MutationTaster, PolyPhen2).

**SP7 S309W knock-in mice showed a complex skeletal phenotype.** To establish causality between the mutation and the skeletal phenotype of our patient, we used CRISPR/Cas9 to generate knock-in mice carrying the S309W *SP7* variant. Three homozygous knock-in mice were generated using this approach, all of which died shortly after birth. Additional founders generated with the same set of CRISPR reagents were confirmed carrying mosaic of S309W variant and short deletions near the target base pair[19], most of which also died neonatally. Four mice mosaic for the S309W variant survived into postnatal life. We performed in vitro fertilization using sperm from a 20-week-old mouse mosaic for the variant to germline transmit the mutant allele, which produced one mouse heterozygous for the S309W allele, but which also died right after birth (Supplementary Fig. 5). The perinatal mortality observed in the mice carrying either the heterozygous or homozygous S309W *SP7* variant may have been due to respiratory failure secondary to the abnormal rib cage with decreased rib length but increased thickness and density, as observed by microcomputed tomography. This mortality limited the number of mice available for study, despite extensive efforts to derive additional mice with the mutation. Overall, we were only able to analyze one heterozygous and three homozygous S309W knock-in newborn mice (Supplementary Data 1). In all four knock-in mice, microcomputed tomography analysis (Fig. 2a–f, also see Supplementary Fig. 6 and Supplementary Movie 1–3) revealed abnormalities in bone shape; for example, the ribs and clavicles (Fig. 2g–i) showed decreased length but increased thickness and density which is reminiscent of the clavicle osteosclerosis in the patient (Fig. 1c). The trabecular density distribution was affected in the long bones of the extremities (Fig. 2j, k) and ribs (Fig. 2d–f), with a marked increase in trabecular bone near the center of the diaphyses, suggesting a failure of medullary cavity formation. In contrast, cortical bone formation appeared impaired (Fig. 2j–l). The homozygous

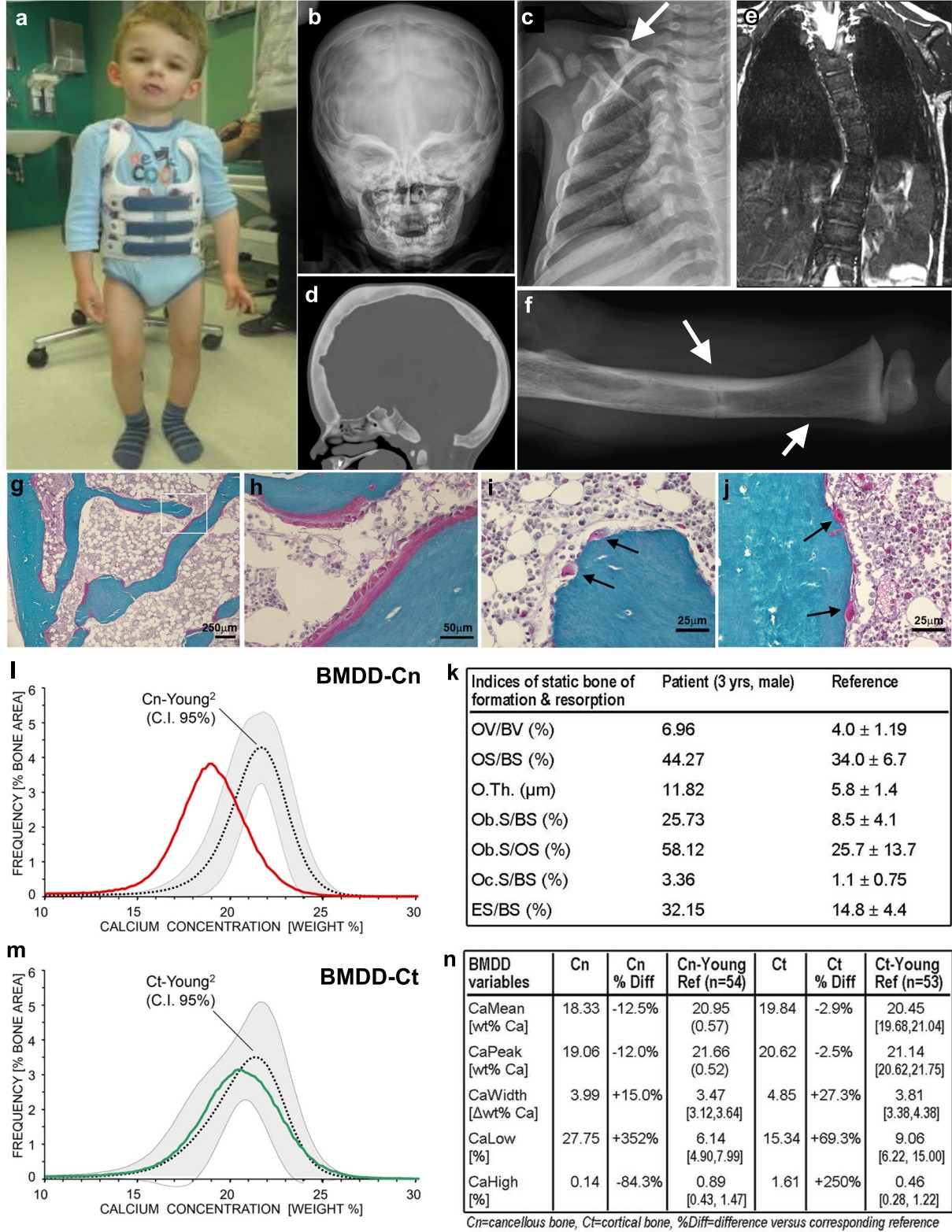

mice showed changes similar to heterozygous mice, but with significantly decreased overall bone density (Fig. 2l, m).

We also performed micro-computed tomography and histological analyses of the four mosaic mice that survived postnatally (two of which died at 24 days and 5 weeks of age, two others were sacrificed at 15 weeks and 20 weeks) (Supplementary Data 2) and found skeletal phenotypes, including thickened clavicle and ribs, uneven cortical thickness, and reduced medullary cavity, similar to the heterozygous and homozygous mutants (Supplementary Fig. 7 and Supplementary Movie 4–7). The observation of a similar phenotype in multiple independent knock-in mice argues against an off-target effect of CRISPR/Cas9 because it would be unlikely for the same off-target change to occur in multiple mice.

We next explored the effects of the S309W mutation on the growth plate by comparing the proximal tibial growth plates of wild-type, heterozygous S309W, and homozygous S309W mice

**Fig. 1 A patient with scoliosis, thickened calvarium, craniosynostosis, heterogeneity of long bone cortical thickness, and multiple long bone fractures due to bone fragility. a** Patient at age 36 months. **b** Skull radiograph showing closure of sutures and scalloping of the inner table, **c** thickened clavicle (arrow), and **d** remarkable thickening of the calvarium. **e** CT scan of spine showed scoliosis at age 2 years. **f** Right femur radiograph showing heterogenous cortical thickness (arrows). Histological section of the transiliac bone biopsy sample showed elevated osteoblast (**g** and **h**, magnified view of rectangle in (**g**)) and osteoclast number (black arrows in **i**, **j**) Goldner staining: green, mineralized matrix; red, unmineralized matrix or osteoid. Ten histological sections of transiliac bone biopsy were prepared and representative images were shown. Static histomorphometric parameters of bone formation and bone resorption compared versus age-matched reference range[41] (**k**). Quantitative backscattered electron imaging showed significantly decreased overall mineralization with increased heterogeneity in both cancellous (**l**) and cortical bone (**m**), compared with pediatric reference range[42]. Red and green curves represent data from the proband, the dotted line indicates median and space between the solid gray lines indicate the 95% confidence interval (C.I.). Cn-Young, cancellous bone; Ct-Young, cortical bone; arithmetic mean of both cortices, respectively (**n**) OV/BV, osteoid volume/bone volume; OS/BS, osteoid surface/bone surface; O.Th., osteoid thickness; Ob.S/BS, osteoblast surface/bone surface; Ob.S/OS, osteoblast surface/osteoid surface; Oc.S/BS, osteoclast surface/bone surface; ES/BS, erosion surface/bone surface[53].

histologically (Supplementary Fig. 8). The histological structure of the cartilaginous growth plates looked similar using Masson's trichrome stain. In situ hybridization suggested a normal distribution of Col10a1, a marker for hypertrophic differentiation. Col1a1 mRNA expression by RNAscope and Sp7 protein expression by immunohistochemistry were detected in the perichondrium and primary spongiosa, but not in growth plate chondrocytes, with similar patterns observed in wild-type mice and those carrying the mutation. Prior studies have detected low-level expression of Col1a1 and Sp7 in growth plate chondrocytes[20–22]. In contrast, the spongiosa, which is the site of endochondral ossification adjacent to the growth plate, showed marked differences between wild-type mice and those carrying the mutation. In the proximal tibial metaphysis, the heterozygous and homozygous mice showed markedly increased extracellular matrix production, by Masson's trichrome staining (Fig. 2o–o"). However, much of this matrix was unmineralized as demonstrated by Gomori's trichrome staining (Fig. 2p, p'). Gomori staining does not provide a quantitative measure of mineral concentration, but the histological finding of decreased mineralized bone is reinforced by the quantitative analysis by micro-computed tomography described above. Together, the results suggest a failure of the mutant osteoblasts to mineralize bone matrix, a finding similar to that seen in the subject's bone biopsy. In the mutant mice, Col1a1 mRNA expression occurred in a disorganized pattern with small groups of expressing cells, as opposed to the normal sheets of osteoblasts lining the regions of cartilage matrix that are being remodeled into bone matrix. The mutant mice also lacked the normal sheets of osteoblasts that normally contribute to formation of the cortical bone[23] (Fig. 2q–q"). Similarly, osteocalcin mRNA expression in the cortex, adjacent to the periosteum[24], was greatly diminished in the mutant mice (Fig. 2t–t"). Unexpectedly, the mutant mice appeared to have increased mRNA expression of Sp7 (Fig. 2r–r").

**S309W mutation is distinct from loss-of-function mutations.** SP7 belongs to the SP/KLF (Kruppel-like Factor) family of zinc finger-containing transcription factors which is required for maturation of pre-osteoblasts into osteoblasts. Consequently, homozygous Sp7-null mice show an arrest in osteoblast maturation, a complete lack of bone formation, and perinatal lethality[1]. Similarly, pathogenic variants in SP7 in humans have been described only in recessive osteogenesis imperfecta (OI type XII) with generalized osteoporosis[7,8,25]. In contrast to both the knockout mice and the prior human subjects, our patient's mutation appeared to be dominant, rather than recessive, and to produce a complex skeletal phenotype with craniosynostosis, increased thickness of the cranium, clavicles, and spine, and uneven thickening of the cortical bone in his lower extremities, not readily explained by simple impairment in osteoblast formation. Similarly, in the knock-in mouse, this mutation was

dominant and showed a complex phenotype, contrasting with the knockout mouse. We therefore hypothesized that the variant in our patient resulted not simply in a loss (or gain) of SP7 function but instead produced a neomorphic function.

**S309W mutant affected osteoblastic differentiation in vitro.** To study the molecular mechanisms by which the SP7 variant found in our subject affects osteoblast function, we infected mouse primary chondrocytes and mesenchymal stem cells (MSCs) with retroviruses expressing green fluorescent protein (GFP, as a negative control and an indicator for successful infection), FLAG-tagged human wild-type SP7 or the FLAG-tagged S309W SP7 mutant. Both wild-type and mutant SP7 protein were expressed at similar levels upon infection (Fig. 3a), suggesting that the mutation did not affect protein translation or stability. Infection with wild-type SP7 increased alkaline phosphatase staining (suggesting enhanced osteoblast differentiation) and Alizarin red staining (Fig. 3b) (indicating enhanced bone matrix mineralization). In contrast, infection with mutant SP7 suppressed both alkaline phosphatase and mineralization, not only compared to wild-type SP7 but also compared to the negative control vector (Fig. 3b). Quantitative RT-PCR showed that mutant SP7 suppressed the expression of genes important for bone matrix mineralization, such as Mmp13, Alpl, Ibsp, and Bglap (Fig. 3c), which was consistent with the observed decrease in alkaline phosphatase and Alizarin red staining. Interestingly, however, we also found that mutant SP7 promoted the downregulation of Sox9 and upregulation of endogenous Sp7 and Col1a1 more so than the wild-type SP7 did (Fig. 3c). The upregulation of Col1a1 was confirmed at the protein level by western blot (Fig. 3a). Thus, the mutant SP7 stimulated expression of some genes involved in osteoblast differentiation but suppressed genes in matrix mineralization. These findings are consistent with the clinical features of our patient with increased osteoblast number but also increased osteoid, indicating defective bone mineralization. However, the decrease in alkaline phosphatase mRNA expression and staining in vitro does not match the increase in serum alkaline phosphatase activity observed in the proband. We can only speculate that the increase in number of osteoblasts in vivo compensates for the decreased expression per cell or that the cell culture systems do not replicate this aspect of the in vivo osteoblast biology. Interestingly, we also found that Rank ligand (Rankl) expression was upregulated by the transfection of mutant SP7 (Fig. 3a and c). This finding suggests that the mutation may similarly increase Rankl expression in vivo, which may stimulate osteoclast differentiation and consequently contribute to the increased bone resorption observed in the patient.

**S309W mutant showed increased GC-box sequence binding.** Next, we investigated the underlying molecular mechanisms that

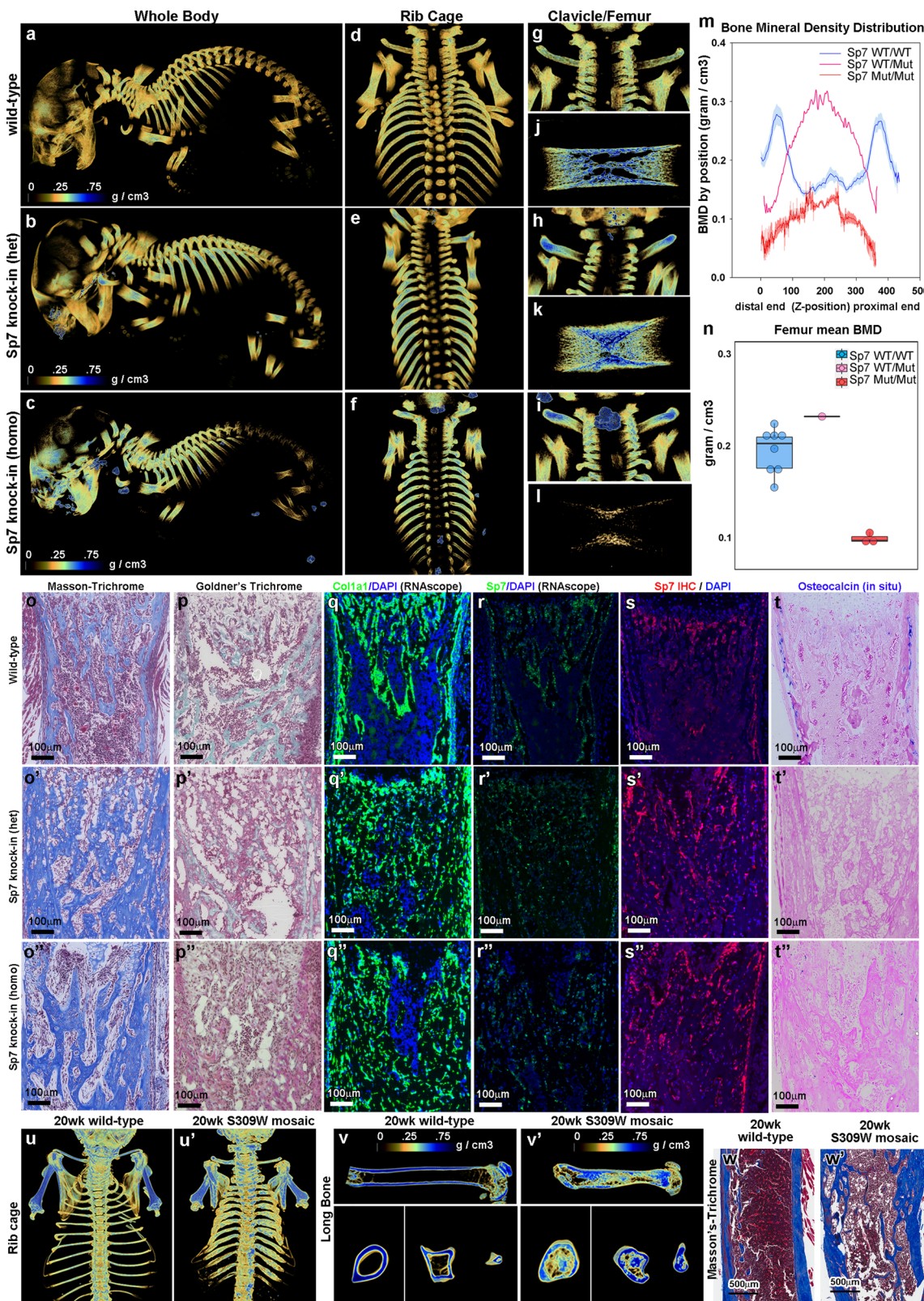

produce this aberrant expression profile. The mutation is located at a highly conserved residue in the first of the three zinc-finger domains (Fig. 4a, Supplementary Fig. 4C), which is conserved across all SP-family members and is involved in binding to GC-box consensus sequences[26–28]. Binding to GC-box is thought to be critical for transcriptional regulation by other SP family members. Although this binding to GC-box has also been

demonstrated in vitro for SP7[1], more recent studies suggest that SP7 may act differently, by recognizing an AT-rich motif through its interaction with DLX proteins[16]. We first investigated whether the mutation affected the binding of SP7 to a GC-box consensus sequence using electrophoretic mobility shift assay (EMSA). As expected, SP1, which is another member of the SP family[29], bound strongly to a GC-box sequence (Fig. 4b, Supplementary

**Fig. 2 Mice with heterozygous S309W SP7 variant showed skeletal abnormalities. a–l** Micro-computed tomography analysis (scale bar indicates relative bone mineral density). Compared to newborn wild-type mice, both S309W SP7 heterozygous and homozygous knock-in mice showed increased bone thickness in the ribs (**d–f**) and clavicles (**g–i**). Long bones in S309W knock-in mice were shortened but increased in diameter (**j–l**), with redistribution of bone mineralization. In the heterozygous knock-in mice, trabecular bone density was decreased at the metaphyses but increased at the diaphyses of the femurs. Homozygous knock-in mice showed a similar pattern but decreased overall bone density (**m**, **n**, $n = 8$ for wild-type, $n = 3$ for homozygous), raw values were included in Source data file. Blue and red lines indicate mean value; light blue and red color bands indicate SEM (**m**). Bold line inside the box represents the median; upper and lower boundary of the boxes represents the 25th and 75th percentile; whiskers represent the 5th and 9th percentile (**n**). **o–t** Histology in the proximal tibial metaphysis showed markedly increased extracellular matrix production in the mutants (**o–o″**) but decreased mineralized bone (**p**, **p′**). Col1a1 (**q–q″**) and Sp7 (**r–r″**, **s–s″**) expression occurred more sporadically and in a disorganized pattern in the mutants and osteocalcin mRNA expression was diminished in the cortex (**t–t″**). Experiment was repeated independently 3 times with similar results and representative images were shown. **u–w** The S309W mosaic mouse used to generate heterozygous S309W knock-in mouse (by in vitro fertilization) was also analyzed by micro-CT imaging at 20 weeks of age (compared to age-matched wild-type). The mosaic mouse similarly showed increased bone thickness in the ribs and clavicles (**u**), shortened but widened long bone with abnormal medullary cavity (**v**). Histology sections showed uneven cortical thickness (**w**). For Masson's-Trichrome and Gomori's Trichrome staining, five histology sections were prepared and representative images were shown.

Fig. S9). In contrast, wild-type SP7 did not significantly interact with the GC-box sequence. This observation is consistent with the recent studies showing that GC-box binding may not be the major mechanism of action for SP7[16]. Interestingly, the mutant SP7 showed modestly increased binding to the GC-box compared to wild-type SP7 (Fig. 4b), which raised the possibility that the mutation may lead to transactivation of genes not normally driven by wild-type SP7. To test this hypothesis, we used retrovirus to express wild-type or mutant SP7 in primary chondrocytes and analyzed binding across the genome, using chromatin immunoprecipitation followed by sequencing (ChIP-Seq). Analysis of binding motifs showed that both wild-type and mutant SP7 recognized GC-enriched sequences (Supplementary Fig. 10), but interestingly the mutant showing increased binding to a large number of genomic regions across the genome that were not bound by wild-type (Fig. 4c). Gene ontology analysis showed that these neomorphic genomic interactions were enriched for genes involved in trabecular bone formation and Wnt signaling (Fig. 4d, Supplementary Fig. 11). For example, mutant SP7 showed increased binding to the promoter region of *Col1a1* and endogenous *Sp7* (Fig. 4e), which may explain the observed upregulation of these genes by mutant SP7 (Fig. 3c). We did not detect binding of either wild-type or mutant SP7 to AT-rich motif in chondrocytes, presumably because the binding to AT-rich motif requires interactions with DLX proteins, which are highly expressed in osteoblasts but not in chondrocytes[16].

**S309W mutant showed decreased AT-rich motif binding**. We next asked whether S309W variant affects its interaction with DLX proteins and AT-rich motif binding. Co-immunoprecipitation showed that both wild-type and mutant SP7 could interact with DLX3 and DLX5 (Fig. 5a). To compare the binding ability of mutant and wild-type SP7, we next performed co-IP experiments in which we expressed both wild-type and mutant SP7 in the presence of DLX3/5. In this situation, we tagged the wild-type SP7 with FLAG-tag and the mutant SP7 with an HA-tag (or vice versa) and let them compete for binding to DLX. We found that, regardless of the tag used, the mutation decreased the ability of SP7 to compete for DLX proteins (Fig. 5b), suggesting that the mutation decreases the affinity of SP7 for DLX3/5. We next performed EMSA and found that DLX5 alone can interact with an AT-rich motif, causing a mobility shift, and confirmed by a supershift induced by antibody against myc-DLX5. Addition of wild-type SP7 caused a second mobility shift, indicating an interaction between DLX5, SP7, and AT-rich oligonucleotides, which was also confirmed by a supershift induced by both anti-FLAG (against SP7) or anti-Myc (against DLX5) antibodies (Fig. 5c and Supplementary Fig. 12). Interestingly, this second shift was not observed when using mutant SP7 (Fig. 5c and

Supplementary Fig. 12). Consistently, a supershift was only observed when using an anti-Myc antibody against the myc-DLX5, but not when using an anti-FLAG antibody against the FLAG-tagged SP7 mutant. We next performed the reverse experiment by altering the concentration of DLX5 in the absence or presence of wild-type and mutant SP7. We found that SP7 alone (wild-type or mutant) did not interact with the AT-rich motif, which explained why the binding motif analysis in ChIP-Seq only identified GC-rich motifs (Supplementary Fig. 10). Increasing the concentration of DLX5 only resulted in a second shift of DNA in the presence of wild-type, not mutant, SP7 (Fig. 5d). We next further assessed the ability of wild-type and mutant SP7 to bind the AT-rich motif by oligo-co-immunoprecipitation. A biotin-labeled oligonucleotide containing the AT-rich motif was allowed to interact with DLX5 and SP7 (either wild-type or mutant), and then co-immunoprecipitation was performed with streptavidin beads, followed by western blot. We found that the mutant SP7 showed diminished, but not completely absent, binding to the AT-rich oligo in the presence of DLX3/5 (Fig. 5e). The findings, together with the EMSA results, suggest that the mutation did not completely abolish but rather decreases the SP7-DLX-AT-motif interaction. Taken together with the competition co-IP results described above, the combined findings indicate that the mutation affects the ability of SP7 to bind DLX proteins and consequently the ability of SP7 to interact, through DLX3/5, with the AT-rich binding motif.

We next performed ChIP-seq in MC3T3 cells transfected with a DLX5-expressing plasmid. Both wild-type and mutant SP7 were able to bind to chromatin, but wild-type SP7 generated more peaks than the mutant (Fig. 5f), suggesting that, in the presence of DLX proteins, the mutation decreases genomic binding of mutant SP7. This finding contrasts with the findings in chondrocytes (which express little DLX), in which mutant SP7 showed increased binding to a large number of genomic regions not bound by wild-type SP7. In these MC3T3 cells, analysis of binding motifs showed that the common peaks and wild-type-specific peaks were very similarly AT-motif enriched sequences (Fig. 5g and Supplementary Fig. 13), while mutant-specific peaks were enriched with sequences different from the shared or wild-type peaks, indicating the mutation alters binding specificity of SP7. This finding also contrasts with the results in chondrocytes where the major binding motifs found were GC-rich sequences. Examples of genomic regions recognized by the wild-type Sp7 but not the mutant includes a region in the Notch2 gene previous reported to bind SP7[16] (Fig. 5h). Taken together, the ChIP-Seq findings in chondrocytes and MC3T3 cells confirm that, in the absence of DLX proteins, wild-type SP7 tends to recognize GC-rich sequences; but in the presence of DLX proteins, wild-type SP7 tends to recognize instead AT-motif sequences. The combined findings also indicate that, in the absence of DLX

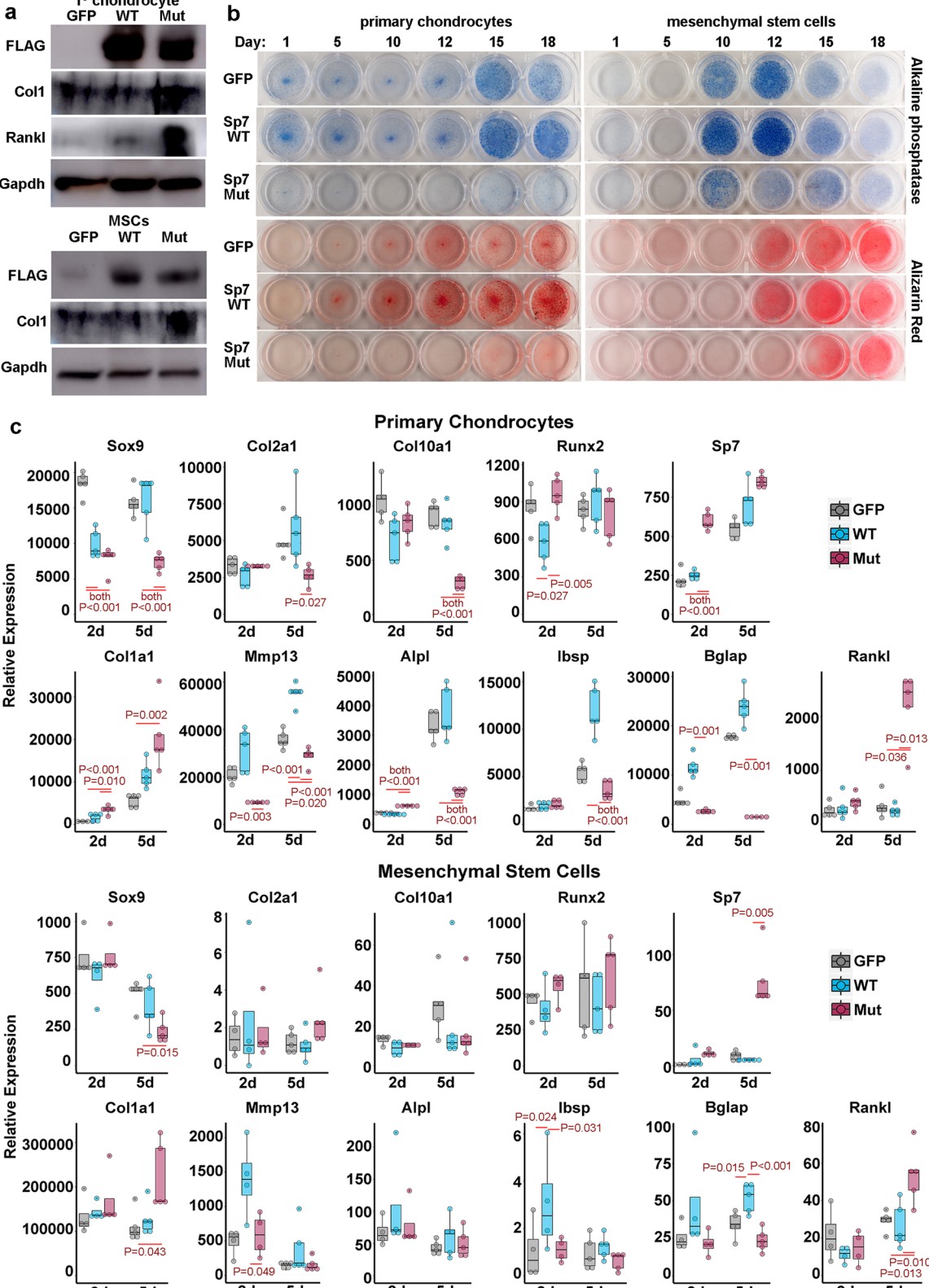

**Fig. 3 S309W SP7 variant affected osteoblastic differentiation in vitro. a** Retroviral expression of wild-type mutant SP7 produced similar protein levels (by detecting FLAG using Western blot) in primary chondrocytes and mesenchymal stem cells (MSCs), but cells transfected with mutant SP7 showed increased Col1 expression and Rankl expression (for chondrocytes) Experiment was repeated independently 3 times with similar results. **b** SP7 variant (versus GFP or wild-type) suppressed normal osteoblast differentiation of primary chondrocytes and MSCs, assessed by alkaline phosphatase and Alizarin red staining. **c** Real-time qPCR showed that infection of mutant SP7 differentially affected expression of genes for osteoblast differentiation or matrix mineralization, shown in box and whisker plots (Rstudio) *$P < 0.05$, ANOVA ($n = 5$), raw values were included in Source data file. Bold line inside the box represents the median; upper and lower boundary of the boxes represents the 25th and 75th percentile; whiskers represent the 5th and 9th percentile.

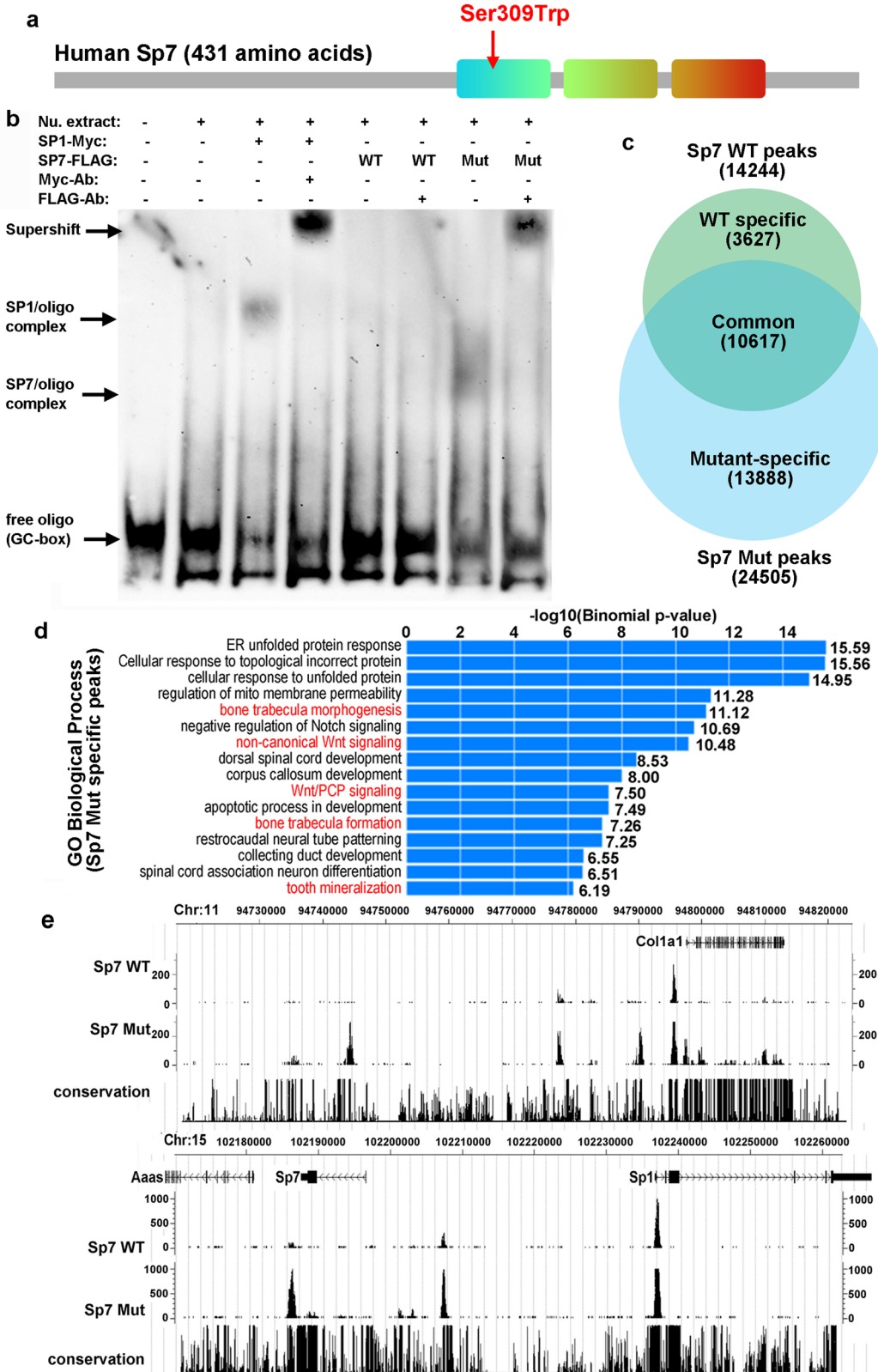

proteins, the S309W mutation increases binding to GC-rich regions, but, in the presence of DLX proteins, the mutation decreases binding to AT-rich motifs, supporting our in vitro binding results from the EMSA and co-IP experiments. These changes in binding properties may provide a mechanism for the observed altered gene expression profile during osteoblastic differentiation.

## Discussion

We studied a boy with a complex skeletal disease which included craniosynostosis, severe scoliosis, long bone fragility, areas of increased bone, particularly with thickened intramembranous bones, and other areas of decreased bone. Histomorphometry revealed increased osteoblast numbers but increased osteoid suggesting impaired osteoblast-mediated mineralization. In this

**Fig. 4 S309W SP7 variant showed increased binding to GC-rich motifs. a** The S309W variant is located in the first of the three zinc finger motifs, which is important for DNA binding. **b** Electrophoretic mobility shift assay (EMSA) in HEK293 showed strong binding of SP1 to a GC-box motif, which was confirmed by myc-antibody-induced supershift. In contrast, wild-type SP7 did not show substantial binding to GC-box motif, while the S309W variant showed increased binding to GC-box motif, confirmed by FLAG-antibody-induced supershift. Experiment was repeated independently 3 times with similar results. **c-e** ChIP-Seq in primary chondrocytes showed increased overall binding (number of peaks) of S309W variant to the mouse genome (**c**). Gene ontology analysis showed that the S309W-specific genomic peaks were enriched with genes involved in trabecular bone formation and Wnt signaling pathway. *P*-values were generated with two-sided test, with Bonferroni correction for multiple testing (**d**). For example, the S309W variant showed increased binding at genomic regions of Col1a1 and Sp7 (**e**).

subject, we identified a pathogenic heterozygous *de novo* missense variant in *SP7*. The complex skeletal phenotype and the apparently dominant nature of the variant differ markedly from the prior cases of SP7-associated recessive osteogenesis imperfecta and is not readily explained by a simple impairment in osteoblast formation due to loss of SP7 function. We created a mouse with the orthologous missense variant in *Sp7*, which produced a complex skeletal phenotype, confirming that the variant is pathogenic, but dissimilar to the *Sp7* knockout mouse, both in inheritance and phenotype, confirming that the variant does not cause a simple loss of function. The human and mouse phenotypes showed some differences. For example, cranial hyperostosis in our patient is not observed in the knock-in mouse. These differences might be due to differences in species and/or the young age of the mice.

We showed evidence that this mutation affects osteoblast differentiation by dysregulating expression of genes important for osteoblast differentiation and matrix mineralization. Furthermore, we showed evidence that the variant decreased binding to DLX proteins and altered the DNA binding specificity of SP7, with increased binding to GC-consensus sequences and decreased binding to AT-rich motif. The variant thus tends to reverse the unique sequence specificity of SP7, causing it to revert to a specificity more similar to the other SP family members. Our current study therefore provides the first in vivo evidence that the unique AT-binding specificity of SP7 is essential for normal bone development.

Very recently, a second subject with the same heterozygous, *de novo* variant (c.926C>G:p.S309W) in *SP7* was reported[30]. This report, which occurred after our subject had been published in abstract form[31,32], included a clinical, radiological, and histological phenotype similar to that of the subject we reported with long bone fragility, scoliosis, thickened calvarium, increased bone turnover and increased osteoid. This case report, however, did not include any experimental evidence that the variant was pathogenic or performed mechanistic investigation but cited our abstract as evidence that the variant was causative. The authors suggested naming the condition juvenile Paget disease-4 because of some shared phenotypic features, including high bone turnover. However, other disorders labeled juvenile Paget disease arise from defects in osteoclast differentiation[33] whereas the SP7-associated disease involves abnormal osteoblast differentiation, and there are also differences in the bone deformities which likely reflect the fundamental differences in pathogenesis. To avoid perpetuating eponym-based nomenclature and to move toward etiology-based nomenclature, we propose the term, SP7-associated high-turnover bone disorder, for this unique phenotype. According to the recently revised classification of genetic skeletal disorders, this new disorder would fall into group 24 (other sclerosing bone disorders), together with, but distinct from, juvenile Paget Disease. A table summarizing the differences between the SP7-associated high-turnover bone disorder described here and other forms of high-turnover disorders was included as Supplementary Data 3.

Many germline loss-of-function variants in transcription factors have been found to cause human disease, as have some gain-of-function variants in transcription factors, such as *STAT1* and *STAT3*[34]. Here we describe a more unusual and novel pathogenic mechanism involving a variant in a transcription factor, *SP7*, which is neither a simple loss-of-function nor gain-of-function but rather a neomorphic mutation. This variant changed the sequence specificity of the transcription factor thereby producing an aberrant transcriptional regulation profile of target genes.

Taken together, our findings suggest that the AT-motif binding specificity of SP7 is physiologically important. Our findings also demonstrate how a neomorphic variant can change the DNA binding sequence specificity of a transcription factor, causing both downregulation of original targets and upregulation of other target genes, resulting in human disease. Our study suggests the possibility that other unresolved rare genetic disorders could be caused by neomorphic mutations in transcriptional regulators, such as transcription factors, histone modifiers, and miRNAs.

## Methods

**Patient report**. The studies described in this paper were conducted according to the principles of the Declaration of Helsinki, with particular attention to §20/vulnerable groups and individuals. Written consent to perform research on material available from an extensive diagnostic work-up was given by the legal guardians of the patient, in accordance with the Ethical Committee of the Medical University of Vienna. In addition, informed consent was obtained from the legal guardians to publish potentially identifying clinical information including the details of the case, radiographs, and photographs that were included in the final version of the manuscript.

The proband is the son of healthy, unrelated parents. The first skeletal symptoms were plagiocephaly and severe lumbar early-onset scoliosis. At the age of 24 months, a femoral fracture after inadequate trauma and a non-recent contralateral femoral fracture was diagnosed. Radiographs revealed bowing of femora and tibiae with cortical anomalies, e.g., sclerotic appearance of the femoral diaphysis with heterogenous lucencies and distended femoral shafts. The spine and clavicles exhibited sclerotic features, which was not observed for the pelvic bones. At age 30 months, progressive macrocephaly was noted, and radiographs showed cranial hyperostosis with scalloping of the inner table involving the orbital roofs and the base of the skull. Neurosurgical interventions included ventriculoperitoneal shunt, craniectomy, and cranial vault remodeling. Sequencing of *TNFRSF11*, *LRP5*, *SERPINF1*, *FGFR1-3*, *TWIST1* did not identify pathogenic variants.

By age 11 years, the boy had experienced 8 femoral, 6 tibial, 2 metatarsal, and 1 sacral fractures. Long bones of the lower limbs were stabilized by intramedullary rods. Treatment with intravenous bisphosphonates was commenced at 4.7 years and led to a substantial decrease in ALP activity and increase in bone mineral content. Linear growth remained within the familial target range despite progressive scoliosis. Under careful clinical, ophthalmologic and neurological follow-up, vision, hearing, linear growth, and gross motor development remained normal.

**Genetic analysis**. The SNP microarray analysis was performed using the Infinium™ OmniExpressExome from Illumina that covers ~950,000 SNPs. Exome sequencing was performed on the proband and both parents at the National Institutes of Health Sequencing Center. A whole-genome library with ~325 base inserts was prepared for each sample using the Kapa DNA Library Preparation Kit (high throughput, with bead) (Kapa Biosystems, Wilmington, MA). Exome capture was performed with the SeqCap EZ Human Exome+UTR Kit v3.0 (Roche Nimblegen, Madison, WI). Each captured exome pool was sequenced on a HiSeq2500 using version 4 chemistry. Data were processed using RTA ver. 1.18.64 and CASAVA 1.8.2. Reads were mapped to National Center for Biotechnology Information (NCBI) build 37 (hg19). All subjects sequencing data were processed in parallel for alignment and variant calling. The combined data were formatted in a single .vs file and analyzed using VarSifter[35] which enables a search for variants in a specific gene and analysis of genotypes using Boolean logic.

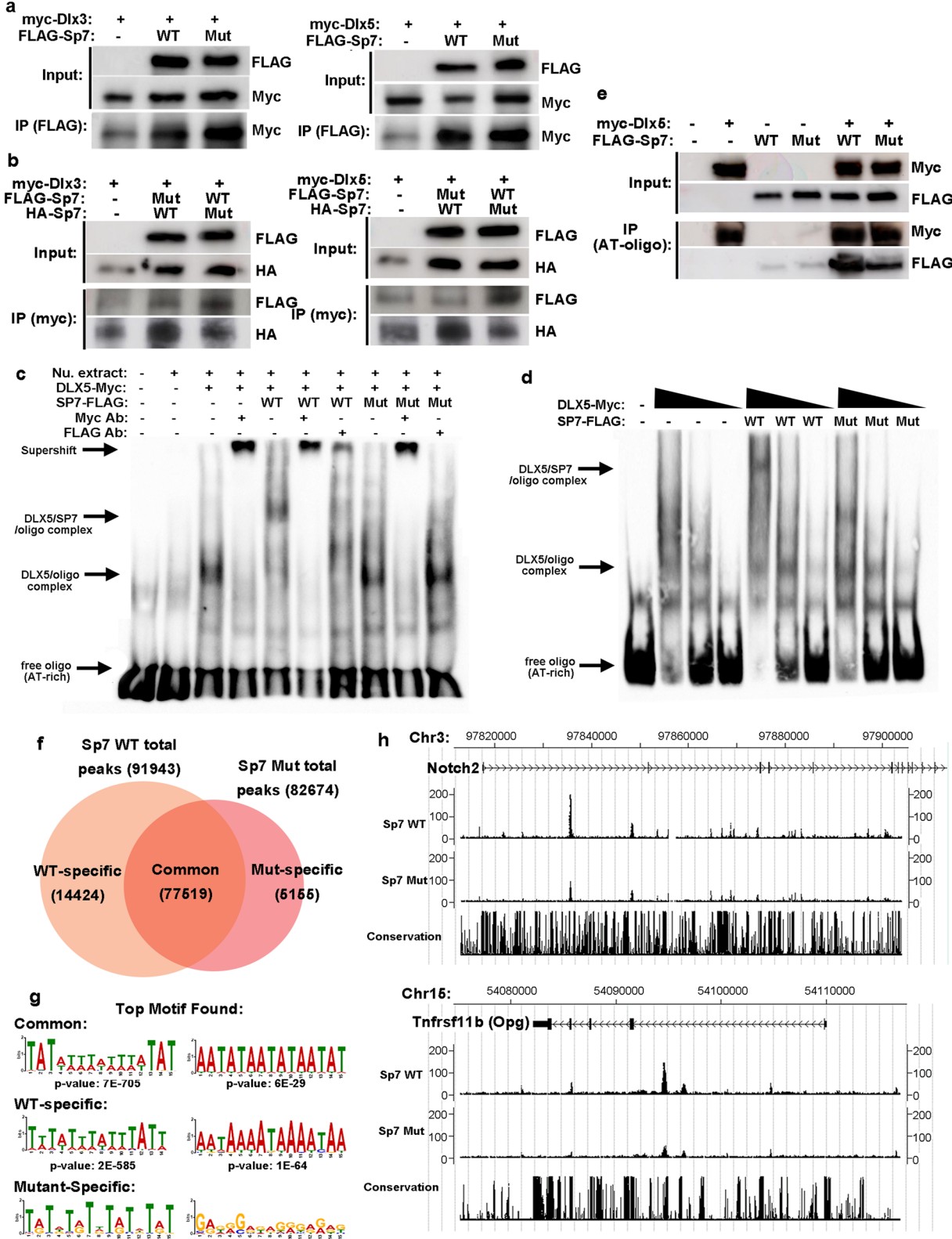

**Animal procedures and generation of Sp7 S309W knock-in**. All animals were used in accordance with the Guide for the Care and Use of Laboratory Animals (National Research Council 2003). The serine to tryptophan substitution at codon 309 was created by CRISPR-mediated genome editing directly in fertilized eggs of C57BL/J (The Jackson Laboratory) as described[36,37]. Target sequences for guide RNAs (gRNAs) near the target codon with minimal off-target hits were initially selected using the ranking tool CRISPR Design (http://crispr.mit.edu) and CRISPRScan.org[38], synthesized with the MegaScript T7 kit (Ambion) and efficacy

confirmed with Surveyor assay. A single-stranded DNA oligonucleotide carrying the SP7 S309W transversion was designed, based upon the position and orientation of the guide RNA, and synthesized (Integrated DNA Technologies). The combination of the guide RNA with the mutagenic oligonucleotide was further tested for its efficiency, introducing the desired point mutation into the mouse genome by digital-droplet PCR with a pair of probes for wild-type and mutated alleles. The final guide RNA selected (GCGGCAAGGTGTACGGCA, 10 ng/µl) was mixed with SpCas9 protein (50 ng/µL; PNA BIO) to form gRNA-Cas9 ribonuclear partials[39],

**Fig. 5 S309W SP7 variant showed decreased binding to AT-rich motifs. a** Co-immunoprecipitation (Co-IP) in HEK293 showed that both wild-type and S309W mutant of SP7 interacts with DLX3 and DLX5. **b** competition Co-IP with HA-tagged wild-type SP7 and FLAG-tagged S309W, or vice versa, showed decreased binding of S309W mutant to DLX3 and DLX5. **c** In EMSA, the S309W variant showed loss of binding to the AT-rich motif in HEK293 cells. In the absence of SP7 (wild-type or S309W variant), DLX5 can interact with the AT-rich motif, confirmed by myc-antibody-induced supershift. Expression of wild-type SP7, but not the S309W variant, induced a second shift of DNA mobility, which is confirmed by both myc antibody and FLAG-antibody-induced supershift, indicating DNA binding activity. **d** Similarly, SP7 (wild-type or S309W variant) alone did not show interaction with the AT-rich motif. An increase in DLX5 expression induced a second shift of DNA mobility, suggesting DNA binding activity in the presence of wild-type but not in the S309W variant. **e** Oligo-co-IP with myc-DLX5 and FLAG-SP7 (wild-type or S309W) showed decreased but not complete loss of binding of S309W to AT-rich oligo probe. All Co-IP, EMSA, and Oligo-co-IP experiment was repeated independently 3 times with similar results. **f–h** ChIP-Seq in MC3T3 cells transfected with DLX5 showed decreased overall binding (number of peaks) of S309W variant to the mouse genome (**f**). Motif analysis identified altered motif specificity of S309W compared to common or wild-type-specific peaks. *P*-values were generated with two-sided test, with Bonferroni correction for multiple testing (**g**). For example, the S309W variant showed decreased binding at genomic regions of Notch2, a known Sp7 target in MC3T3 cells, and Opg (**h**).

and the mutagenic oligo (127mer, GTGAACCTCTTGCCGCAGAAAAGCCAG TTGCAGAC GAAAGGCCTCTCGCCAGTGTGCCAGCGCAAGTGGGCTTTC AGATGCcAAGCtTTtCCGTACACCTTGCCGCACCCAGGGATGTGGCA GCTGT, 100 ng/µL). The mixture was microinjected into fertilized eggs isolated from the C57BL6/J donor females (Jackson) to generate the knock-in allele. Injection dosage of the gRNA was titrated carefully to the minimal effective concentration, 1 ng/ul in this case, in a gRNA screening assay involving electroporation of a range of dosages (0.1–50 ng/ul) of the gRNA. With these precautions, the off-target effects are expected to be minimal[40]. Genomic DNA was obtained from tail snips of F0 mice. A PCR amplicon spanning the mutation was generated and TA-cloned (Invitrogen) and Sanger sequenced. The sequence of primers used for genotyping is as follows: forward, CCAAGGCGGTTGGCAATAGT; reverse, CTCAAGTGGTCGCTTCTGGTAAAG. At least 10 clones were sequenced for each animal. Mice containing only the correct mutation without nonsynonymous mutation in all clones were considered homozygotes. One mosaic mouse that has the S309W mutation was used for in vitro fertilization to generate germline transmitted heterozygous mouse. A summary of age, sex, and genotype used in the current study is summarized in Supplementary Data S2.

**Bone histomorphometry and quantitative backscattered electron imaging (qBEI)**. The patient had a transiliac bone biopsy performed following double-labeling with tetracycline to allow dynamic measurement of bone formation. Sample preparation and histomorphometric analyses were performed using standard procedures and results were compared to reference data of healthy age-matched controls)[41]. Subsequently, bone mineralization density distribution (BMDD), reflecting the calcium content of cortical and trabecular bone matrix, was measured by qBEI using a digital electron microscope (DSM 962; Zeiss Oberkochen, Germany) equipped with a four-quadrant semiconductor backscattered electron BE detector[17]. The entire cross-sectional area was scanned and BMDD in trabecular and cortical bone compared to controls as described previously[42].

**Micro-CT**. To study bone mineral density and bone structure, animals were fixed directly in 10% formalin for at least 3 days at 4 °C and stored in phosphate-buffered saline. Postnatal animals were fixed in 10% formalin for at least 2 weeks. µCT scanning was performed as previously described[43] using the following settings. For newborn mice, a SkyScan 1172 (Skyscan, Kontich, Belgium) was used at 13 µm resolution for whole body and at 5 µm resolution for individual long bones. For 24-day-old mice, a SkyScan 1272 was used at 6 µm resolution for individual long bone, and Quantum GX (PerkinElmer) was used at 72 µm resolution for whole-body scans. For 5-wk-old mice, the SkyScan 1172 was used at 5 µm. For 15-wk-old mice, a Quantum GX was used at 72 µm resolution. For 20-wk-old mice, a SkyScan 116 was used at 17.5 µm resolution. 3-Dimensional reconstruction of scanned datasets was performed using NRecon (Bruker, Billerica, MA) and rotated (only for long bones) with DataViewer (Bruker). Whole-body µCT images and videos were generated by CTVox (Bruker). Morphometric analysis of trabecular and cortical bone was performed according to method notes MN094 and MN095 (Bruker user manual), respectively, to obtain measurements of positional bone mineral density (BMD).

**Bone and growth plate histology**. Histological evaluations were performed on proximal tibia and distal femur sections stained with modified Gomori's-stain (for undecalcified bone) or Masson Trichrome stained (for decalcified bone) and visualized using a Keyence BZ-X700 fluorescence microscope (Keyence Corp, Osaka, Japan) at ×10 magnification under bright field.

**In situ hybridization**. In situ hybridization was performed as described previously[43]. Briefly, riboprobes for collagen X and for Osteocalcin[44] were generated by PCR using mouse growth plate cDNA as template and primers that contained an SP6 promoter. Single-stranded digoxigenin-labeled riboprobe for in situ hybridization was transcribed using a DIG RNA Labeling Kit (Roche Diagnostics) following the manufacturer's protocol. Riboprobes were purified by Micro Bio-Spin

Columns P-30 Tris RNase free (Bio-Rad). Paraffin-embedded sections of newborn bone/growth plate were hybridized to digoxigenin-labeled riboprobes (100 ng riboprobe per slide or 50 ng per section). Antigen retrieval was performed by proteinase K incubation (10 µg/ml, 37 °C for 30 min). For detection, tissue sections were incubated with anti-digoxigenin alkaline phosphatase Fab fragments (Roche) for 2 h at room temperature and treated with NBT/BCIP (Sigma) in the dark until a colorimetric change was detected (1 h for Collagen X; overnight for Osteocalcin). Sections were counterstained with 10% eosin and visualized using Keyence BZ-X700 fluorescence microscope (Keyence Corp, Osaka, Japan) at ×10 magnification under bright field.

**RNAscope**. Paraffin-embedded sections of newborn bone/growth plate were processed following manufacturer's standard protocol (Advanced Cell Diagnostics Inc., Newark, CA). Briefly, slides were baked at 65 °C for 1 h, followed by deparaffinization. Antigen retrieval was performed by heating the slides in antigen retrieval buffer (provided by manufacturer) to 99 °C for 20 min, followed by protease plus incubation at 40 °C for 30 min. Slides were incubated with probes for Col1a1 (Mm-Col1a1, 319371) or Sp7 (Mm-Sp7-C2, 403401-C2) for 2 h at 40 °C, followed by signal amplification and detection using the RNAscope Multiplex Fluorescent Reagent Kit version 2. Opal 520 or Opal 690 fluorophores (AKOYA Biosciences, Marlborough, MA) were used for signal detection at 1:1000 dilution in TSA buffer. Sections were counterstained with DAPI (300 nM) and mounted with ProLong Diamond Antifade Mountant (Thermo Fisher Scientific) and visualized using Keyence BZ-X700 fluorescence microscope.

**Immunohistochemistry (IHC)**. Paraffin-embedded sections of newborn bone/growth plate were baked at 65 °C for 1 h, deparaffinized in xylene, rehydrated through ethanol series (100%, 100%, 95%, 70%), and rinsed with PBS. Antigen retrieval was performed by heating the slides in citrate buffer (pH6.0) to 99 °C for 15 min. Endogenous peroxidase activity was blocked by 3% $H_2O_2$. Sections were incubated with anti-Sp7 (ab22552, Abcam) 1:500 dilution in 10% goat serum (Vector Laboratories, Burlingame, CA) overnight at 4 °C, followed by Alexa-Fluor647 goat anti-rabbit secondary antibody (Thermo Fisher Scientific). Sections were counterstained with DAPI (300 nM) and mounted with ProLong Diamond Antifade Mountant (Thermo Fisher Scientific) and visualized using Keyence BZ-X700 fluorescence microscope.

**Plasmid preparation**. Plasmids expressing GFP, FLAG-tagged wild-type or mutant SP7, and myc-tagged SP1 were constructed by Q5 High-Fidelity DNA Polymerase (New England Biolabs) using a pMX retroviral vector (for MSCs and chondrocytes) or pcDNA3.1 vector (for HEK293 cells). Plasmid expressing myc-tagged DLX3 or DLX5 was constructed by gene synthesis (Gene Immune) using a pcDNA3.1 vector.

**Chondrocyte and MSC isolation and culture**. Growth plates from proximal tibias and distal femurs were dissected from 1-week-old wild-type C57BL/6 mice aseptically and digested in 0.3% collagenase type I (Sigma-Aldrich) in Dulbecco's modified Eagle medium (DMEM)/F12 medium. The released cells were resuspended, and cultured at a starting concentration of 0.03 million per cm² in DMEM/F12 medium (Invitrogen) supplemented with 10% fetal bovine serum, penicillin (100 U/mL)/streptomycin (100 µg/mL), and 50 µg/mL ascorbic acid in a humidified incubator at 37 °C, 5% CO₂. Mouse whole femurs, rather than bone marrow tissues, were used to isolate MSCs as previously described[45], because hematopoietic cells continue to exist in bone-marrow-derived MSC cultures even after many passages[46]. To obtain MSCs, 3- to 5-week-old C57BL/6 mice were killed and briefly immersed in 70% (vol/vol) ethanol for 5 min. Femurs were isolated from both legs. The muscles and tendons were carefully removed using scalpel blades. Bone marrow was flushed out of the bones and the bones were minced with heavy-duty scissors (18054-11, Fine Science Tools, Foster City, CA). Remaining soft tissues were removed with suction while the bone chips were being washed with PBS. MSCs were liberated by digesting the bone chips with collagenase D

(Roche Diagnostics, Indianapolis, IN). The MSCs together with the partially digested bone fragments were cultured in 25-cm cell culture flasks with MSC basal media (Stem Cell Technologies, Vancouver, Canada) supplemented with MSC stimulatory supplement (Stem Cell Technologies, Vancouver, Canada) until it was passaged four times or two weeks before being used for retroviral infection.

**Chondrocyte and MSC retroviral infection.** Three days prior to infection. Phoenix Ecotropic (ECO) cells (obtained from ATCC, Manassas, VA) were transfected with pMX retroviral plasmids (GFP, wild-type SP7, or mutant SP7) using lipofectamine 2000 according to manufacturer's standard protocol. ECO cell culture media were refreshed on the next day, and the day before the infection. On the day of infection, the retrovirus-containing ECO cell culture media were collected and filtered with a 0.45 μm membrane filter (Millipore), mixed 1:1 with full chondrocyte or MSC culture media, and supplemented with 8 μg/ml polybrene (Sigma-Aldrich). Chondrocytes or MSCs were infected twice with the media (once in the morning and second time overnight or longer). Successful retroviral infection was indicated by GFP signal in cells after incubation (close to 100% infection efficiency).

**MC3T3 cell culture and transfection.** MC3T3-E1 cells (subclone 4) were obtained from ATCC (Manassas, VA) and maintained in alpha-MEM medium (Invitrogen) supplemented with 10% fetal bovine serum, penicillin (100 U/mL)/ streptomycin (100 μg/mL) in a humidified incubator at 37 °C, 5% $CO_2$. MC3T3 cells were transfected with pcDNA3.1 expressing FLAG-tagged wild-type or mutant SP7 using lipofectamine 2000 (Invitrogen) according to manufacturer's standard protocol. Cells were collected 48 h post-transfection for ChIP-Seq.

**Western blot.** Total protein was isolated from cultured primary chondrocytes, MSCs, or HEK293 cells after transfection using RIPA buffer supplemented with proteinase inhibitor cocktail (Sigma-Aldrich). Western blotting was performed as previously described[47], using anti-FLAG tag (M2, Sigma-Aldrich), anti-myc tag (TA150121, Origene), anti-HA (ab18181, Abcam), anti-Col1a1 (ab34710, Abcam), anti-Rankl (ab45039, Abcam), or anti-Gapdh (ab9485, Abcam) antibodies. All uncropped images were provided in the Supplementary data.

**Osteoblast differentiation and assay.** Chondrocyte maturation/differentiation was induced by supplementing the chondrocyte culture medium with 50 μg/ml ascorbic acid and 10 mM β-glycerophosphate (Signa-Aldrich) as previously described[48]. Osteoblastic differentiation of MSCs was induced by osteogenic media (CCM 009, R&D System, Minneapolis, MN). Osteoblast differentiation was evaluated with alkaline phosphatase staining and Alizarin Red staining as described[48]. Briefly, after differentiation, cells in 24-well culture plates were fixed in 10% formalin, washed with PBS, and incubated with Alkaline phosphatase staining solution (Thermo Scientific) or 1% Alizarin Red (vol/vol) in 95% ethanol for 5 min at 37 °C.

**RNA extraction and quantitative real-time PCR.** RNA was extracted using RNeasy Micro Kit (QIAGEN, Valencia, CA). All RNA samples had a 260/280 nm ratio between 1.8 and 2.1. RNA integrity was determined using an Agilent 2100 Bioanalyzer (Agilent Technologies, Santa Clara, CA), and only high-quality RNA (RIN > 8) was used for quantitative real-time PCR. Total RNA (500 ng) was reverse-transcribed using SuperScript IV Reverse Transcriptase (Invitrogen). Real-time qPCR was performed as previously described[47] using commercially available FAM- or VIC-labeled Taqman assays (Applied Biosystems, Foster City, CA). The assays used in the current study includes Sp7 (Mm04209856_m1), Col1a1 (Mm00801666_g1), Col2a1 (Mm01309565_m1), Col10a1 (Mm00487041_m1), Sox9 (Mm00448840_m1), Runx2 (Mm00501584_m1), Alpl (Mm00475834_m1), Ibsp (Mm00492555_m1), Mmp13 (Mm00439491_m1), Bglap (Mm03413826_mH), Rankl (Mm00441906_m1), 18S rRNA (4352930E). In particular, the Sp7 probe chosen amplifies a region spanning exon 1 and 2 of the mouse Sp7 mRNA. The coding region of Sp7 mRNA starts in exon 2, and because the retrovirus constructs we used to express human wild-type and mutant SP7 start at the coding region of human exon2, they do not contain sequence in exon 1 and thus will not be amplified by the Sp7 taqman assay. Reactions were performed in triplicate on cDNA derived from each animal using the ABI QuantStudio 6 Flex System instrument (Applied Biosystems). The relative quantity of each mRNA was calculated using the formula relative expression $= 2^{-\Delta Ct} \times 10^6$, where Ct represents the threshold cycle and $\Delta Ct = (Ct$ of gene of interest$) - (Ct$ of 18S rRNA$)$. Values were multiplied by $10^6$ for convenience of comparison.

**Co-immunoprecipitation.** HEK293 cells were obtained from ATCC (Manassas, VA) and were maintained in DMEM medium (Invitrogen) supplemented with 10% fetal bovine serum and penicillin (100 U/mL)/streptomycin (100 μg/mL) at 37 °C, 5% $CO_2$. For co-immunoprecipitation (co-IP), cells were co-transfected with pcDNA3.1, or pcDNA3.1 expressing FLAG-tagged SP7 (wild-type or mutant) and pcDNA3.1 expressing myc-tagged DLX3 or DLX5. Forty-eight hours post-transfection, protein was extracted for co-IP using anti-FLAG antibody (M2, Sigma-Aldrich) and the Dynabeads Protein G Immunoprecipitation Kit (Invitrogen) according to manufacturer's protocol. Lysate before co-IP (as input) and eluted

protein were assessed by western blot. For competition co-IP, cells were co-transfected with pcDNA3.1 expressing myc-tagged DLX3 or DLX5, and an equal amount of pcDNA3.1 expressing HA-tagged wild-type SP7 and FLAG-tagged mutant SP7, or HA-tagged mutant SP7 and FLAG-tagged wild-type SP7. Co-IP was performed using anti-myc (ab-9106, Abcam) and eluted proteins were assessed by western blot with anti-FLAG (M2, Sigma-Aldrich) and anti-HA (ab18181, Abcam).

**Electrophoresis mobility shift assay (EMSA).** Forty-eight hours post-transfection with plasmids expressing SP1, SP7 (wild-type or mutant) and/or DLX5, nuclear protein extraction was performed by lysing HEK293 cells with cytosolic extraction buffer (10 mM HEPES-KOH pH 7.8, 10 mM KCL, 0.1 mM EDTA pH 8.0, 0.1% NP-40), followed by nuclear extract buffer (50 mM HEPES-KOH pH 7.8, 420 mM KCL, 0.1 mM EDTA pH 8.0, 5 mM $MgCl_2$, 20% Glycerol). EMSA was performed using a Light Shift Chemiluminescent EMSA Kit (20148; Fisher Scientific) with 2 μg of nuclear extract, following the manufacturer's instructions, except that 12% glycerol was used in the reaction. For supershift assays, the EMSA reaction minus the oligo probe were first incubated with 1 μg of anti-myc or anti-FLAG antibody for 30 min on ice, followed by a further 30 min incubation on ice in the presence of oligo probe. DNA probes[16] labeled with biotin were synthesized by Integrated DNA Technologies. GC-box oligo probe sequence: GTTGCCAGAAGCCCCGCCCCTAGGAGTGATCGGAAAG; AT-rich oligo probe sequence:ATGGTAATTATAGGTTGGCATAATTATAGGTTTCGATAA TTATAGGTTTCAC.

**Oligo co-immunoprecipitation (oligo co-IP).** Two hundred micrograms of nuclear extract from transfected HEK293 cells was used for each reaction of oligo co-IP. Reaction buffer contained 50 mM KCl, 25 mM HEPES-KOH (pH 7.7), 5 mM $MgCl_2$, 50 μM Zn(OAc)$_2$, 1 mM DTT, 12% glycerol, 0.1% NP-40, 1 μg poly(dI-dC), 100 pmol biotinylated AT-rich oligo probe (same sequence as used in EMSA). Nuclear extract was incubated with biotin-oligo and streptavidin magnetic particles (Sigma-Aldrich) at 4 °C for 2 h, followed by three washes with phosphate-buffered saline on a magnetic stand, eluted in Lane Marker Reducing Sample Buffer (Pierce) and RIPA buffer. Lysate before co-IP (as input) and eluted protein were assessed by western blot.

**ChIP-seq and data analysis.** ChIP was performed as previously described[16]. The construction of ChIP-seq libraries was performed with a SMARTer ThruPLEX DNA-seq Kit (R400428, Takara Bio) according to the manufacturer's instructions. The library was sequenced on Hiseq X (Illumina) platforms. DNA-sequence information was aligned to the unmasked mouse genome reference sequence mm9 by bowtie aligner[49]. Peak calling was performed by two-sample analysis on Cis-Genome software[50] with a $p$-value cutoff of $10^{-5}$ comparing with the input control without antibody immunoprecipitation. Peaks were incorporated into further analysis displaying an FDR < 0.01. For peak distribution in genome and gene ontology analysis, GREAT GO analysis was performed utilizing the online GREAT GO program, version 4.0.4[51]. Each peak category was run against a whole-genome background with assembly mm9. De novo motif analyses were performed using DREME[52]; a 100 bp region surrounding peak center was extracted from mm9 and used for the analysis. Peak intersection was performed by BEDTools-Version-2.16.2. The ChIP-Seq raw data were deposited in GEO database with the accession number GSE151217 (for chondrocytes) and GSE180367 (for MC3T3).

**Statistical analysis.** The exact sample size ($n$) for each experimental group/condition, wherever applicable, was provided in the figure legends. All distinct data points shown in the paper were taken from distinct samples rather than from same sample measured repeatedly. For analysis of gene expression in real-time PCR, two-sided one-way ANOVA was performed (passed normality test), corrected for multiple comparisons (Bonferroni test). When normality test failed, two-sided one-way ANOVA was performed on ranks, corrected for multiple comparisons with Tukey method. For boxplots, A line is drawn inside the box representing the median (the 50th percentile), while the upper and lower boundary of the boxes represents the 25th and 75th percentile. Whiskers represent the 5th and 9th percentile. Our current study did not involve covariation analysis or Bayesian analysis.

**Reporting summary.** Further information on research design is available in the Nature Research Reporting Summary linked to this article.

## Data availability

All data generated or analyzed during this study, including the raw values of dot pots and scatter plots, and uncropped images of western blot, are included in this published article (and its supplementary information files). ChIP-Seq data were available in GEO database with the accession numbers GSE151217 (chondrocytes) and GSE180367 (MC3T3). Source data are provided with this paper.

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

## Acknowledgements

This work was supported by the Intramural Research Program of the Eunice Kennedy Shriver National Institute of Child Health and Human Development, NIH; the AUVA (Research funds of the Austrian workers' compensation board); and the OEGK (Austrian Health Insurance Fund). The authors thank the patient and his family for their consent and support of research studies. The authors thank Danielle Donahue from the NIH mouse imaging facility for providing training and technical support on mouse micro-CT and data analysis. The expert orthopedic surgical care (Prof. Rainer Biedermann, University of Innsbruck, Austria) and neurosurgical care (Dr. Andrea Reinprecht, University of Vienna, Austria) are very much appreciated. The authors also thank Prof. Michael Rasse from University of Innsbruck for performing a complicated skull operation in the patient to create more space for the brain. The authors also wish to acknowledge the local

medical team at Braunau Children's Hospital, in particular Dr. Sonja Szekely and colleagues for their continuous and dedicated pediatric care and Dr. Robert Stelzl for radiological diagnostic work-up. The authors thank Prof. Klaus Klaushofer from Ludwig Boltzmann-Institute of Osteology for support and discussions. The authors also thank Dr. Steven Coon, Dr. James Iben, and Dr. Ryan Dale from Eunice Kennedy Shriver National Institute of Child Health and Human Development for help with data analysis of ChIP-Seq in MC3T3 cells, and Mr. James L. Herrick and Dr. Theresa E. Hefferan from Biomaterials and Histomorphometry Core of Mayo Clinic (Rochester, MN) for help with sectioning and staining of mineralized mouse bone samples.

## Author contributions

J.C.L. designed, performed, and interpreted experimental studies of SP7 function in vitro and in vivo. J.B. contributed to the overall study design and data interpretation. Y.H.J. designed some experimental studies and analyzed SNP array and exome sequencing data. L.D. created the mouse model. H.H. performed the chromatin immunoprecipitation-RNAseq experiments. B.K. performed experimental studies of SP7 function in vitro. P.R. and N.F.Z. analyzed and interpreted patient bone biopsy sample. G.H. and A.R. initiated pathophysiological concepts and coordinated collaborations. U.W., G.H., and A.R. contributed clinical data and clinical care of the patient. J.C.L. and J.B. wrote the manuscript with contributions from Y.H.J., A.R., N.F.Z., and G.H.

## Funding

## Competing interests

The authors declare no competing interests.
