## [Peer Review File · Nature Communications]

Reviewers' Comments:

Reviewer #1:

Remarks to the Author:

Julian Lui et al. present an interesting manuscript on the role of a novel mutation in SP7/Osterix gene on expression of bone related genes and bone formation in vivo and in vitro. The authors present a set of results implicating a role of mutant SP7/Osx on skeletal development and osteogenesis in humans and mouse. The manuscript tries to provide insightful results including a possible pathological involvement of a sequence specificity modification of zinc-finger motifs. While this concept would draw attention from several different communities, a few issues need further consideration to publish in nature communications.

Major comments

1. It is not clear whether an Austrian boy is homozygous, heterozygous or mosaic for the mutation. Presentation of genomic sequence data will be helpful for readers to evaluate phenotypes of the patient and mutant mice.
2. Heterozygous *Osx* mutants were reported to be normal in general features. Formation of the rib cages in the S309W hets seems to be unaffected in microCT images in fig.2, although they showed delayed bone formation. Therefore, a cause of very low survival rates and very low recovery rates of pups of mutant mice is unclear. The authors should describe the reasons. In addition, this reviewer assumes that this very low recovery rates of mutant pups gave the authors difficulties to analyze phenotypes. As shown in ext. fig.5, production of mutant mice with high mosaicism should be taken into account for further detailed examination including histological studies in postnatal stages. This would also open the door to perform bone histomorphometry in mutant mice.
3. Since generation of mutant lines and a clear description of their phenotypes are requisite to prove an S309W mutation in *Osx* is a cause of the disorder, a thorough examination of the mutant mice is a core of this manuscript. However, only microCT images and bone mineral density distribution are presented. Histological examination of bones formed through intramembranous and endochondral ossification is necessary. Especially, histological studies of growth plate cartilage are required for following reasons. *Osx* is expressed in the mature zone of the mouse growth plate. Expression level of *Col1a1* was increased in growth plate cartilage when *Sox9* was deleted in tissue specific manner (Peter Dy et al., *Developmental Cell* 22, 597–609, 2012). Recent lineage tracing experiments showed growth plate chondrocytes differentiated into osteoblasts in mouse early development (Koji Mizuhashi et al., *Nature*, 563, 254-258, 2018). From these findings, it is reasonable to hypothesize *Osx* in the growth plate has some roles in cartilage to promote endochondral ossification, and hence mutation could affect it. Indeed, the microCT image of the S309W mutant long bone in fig.2K showed delayed ossification (a distance between primary ossification centers are short) in the long bone. Are there any morphological irregularities in growth plate structures? In addition, did mutants show any dysregulation of gene expression patterns of chondrocyte specific genes and aberrant activation of mutant *Osx* and *Col1a1* expression in growth plates? Histological examination and in situ hybridization studies on mutant mice growth plates and primary ossification centers are necessary.
4. The authors tried to explain the possible functional alteration of mutant *Osx* by introducing the recent idea of possible association of *Osx* with *Dlx*s. This reviewer is neutral to this concept, however, data presented in fig. 4 is not sufficient enough to reach solid conclusion. First, fig.4B showed relative affinity/binding activities of Sp1 and *Osx*. The authors used exogenous Sp1 and *Osx* proteins to show band shifts. These results are perplexing. The authors transfected recombinant genes by retrovirus into HEK cells and preparing nuclear extracts for EMSA. If I did not misunderstand, their nuclear extract is rich in endogenous Sp proteins which could form strong signals in the EMSA. However, a shifted band is only observed in the mixture with SP1-myc nuclear extracts. In addition, the mixture with SP1-myc nuclear extracts showed no free probe left in the bottom of the lane, even though Sp7-flag lane showing sifted signals left descent amounts of free probes. Identities of sifted complexes are not clear. Second, fig.4G and H are critical for the authors to describe "in vivo evidence in both mouse and humans that the AT-motif binding specificity of SP7 is physiologically important" in the line 222 and 223. However, again, identities

of sifted complexes are not clear (competition with cold probes is sufficient enough). Third, IP experiments showed association of Osx and Dlx in the absence of DNA fragments and mutants showed relatively strong binding to Osx, suggesting association of wt/mut Osx and Dlx is DNA-binding independent. However, In EMSA, mutant proteins had less activity for complex formation. It is difficult to reconcile these controversial results.

5. ChIP-seq analyses of occupancies of recombinant proteins showed mutant proteins are enriched in the genomic regions where wild type Osx occupancies are low, suggesting that mutation provides changes sequence preferences. These results were obtained from transfected cells in which Dlx are not highly expressed. Is it possible to show that mutant proteins are not enriched in the genomic regions where Dlx occupancies are high in Dlx expressing cells?

6. The authors proposed that S309W mutation caused a high turnover skeletal disorder. However, mechanisms were not described in this manuscript. Even in the in vitro exp in fig.3, effects of mutant proteins on RANKL mRNA expression, Osx protein expression, and type I collagen expression were not analyzed. The authors should present and discuss the possible mechanisms of a high turnover skeletal phenotype.

7. This reviewer noticed that the patient shows normal linear growth. Nevertheless, as described in comments above, mutant phenotypes including human symptoms could be partially explained by "changes of sequence preferences to GC-rich genomic regions affect chondrocytes differentiation in growth plate". The discussion in the line 222 and 223 is a bit aggressive when it comes to mechanistic implication of mutant Osx during development.

Minor comments

1. The features of a complex skeletal disorders of the patient are documented to a certain extent. It would be helpful for readers to summarize differences between the Osx-associated high turnover disorder and other form of high turnover disorders in a table.

Reviewer #2:

Remarks to the Author:

The goal of this study was to identify and characterize the mutation responsible for a skeletal disorder in a young boy. The authors identified a missense mutation in the Sp7 transcription factor and found that it caused altered DNA binding and gene activation. Overall, the studies are comprehensive and convincing.

Comments.

1. The molecular basis for the increase in bone resorption in the human patient is unclear. Because the mutant mice die so early, it is unlikely that they could be analyzed for such mechanisms. However, expression of the mutant proteins in cells that support osteoclast formation might provide some insight. Alternatively, the authors may possess some chromatin immunoprecipitation results that might shed light on this issue.

2. The legend for figure 2 lacks sufficient detail. A-L appear to be microCT images, but this needs to be stated. Also, it is unclear what the color-coding represents (mineralization?). In panel M it is not clear which bone was analyzed and in panel N it is unclear what genotype is represented by the box and whiskers (wild-type, sample number?)

3. In Extended Data element 2, it is unclear what the red boxes indicate.

Reviewer #3:

Remarks to the Author:

The manuscript by Lui JC et al describes a de novo missense mutation (p.S309W) in the

transcription factor SP7, in a patient with a bone phenotype characterized by craniosynostosis, osteosclerosis of the clavicles and spine and bone fragility in the lower extremities. The same mutation has been previously reported as the cause of the disease in another patient with similar phenotype (Whyte MP et al. Bone 2020), thus supporting causality. The authors knocked-in the mutation in mice by CRISPR/Cas9 and observed a neonatal lethal phenotype for both homozygous and heterozygous S309W mutants. The knock-in mice are reported with skeletal features that remind the human phenotype. The authors also performed molecular studies in cultured cells that associate the mutation with a neomorphic function of SP7 due to increased GC-box binding and decreased binding to AT-rich motifs. This work validates the pathogenicity of the S309W mutation, and interestingly, sheds light on the underlying pathomechanism. However, there are parts of the manuscript that need to be more thoroughly explained and experimental procedures that should be revised to strengthen the results.

Comments:

1. Patient Data

1.1. The patient of this manuscript and the patient reported by Whyte MP et al., both have a marked elevation of ALP. In addition, the patient described by Whyte MP et al. also had increased urinary hydroxyproline but normal deoxypyridinoline. The authors could also check these two parameters in urine to help drawing the biochemical picture of this condition. Classical DXA analysis is a helpful parameter for characterization of bone disease and should be added to the patient's clinical description.

1.2 The authors indicate that "bone histomorphometry showed increased osteoblast and osteoclast numbers (Figure 1G-J)". The type of studies and data (whether it was cell counting/bone surface or else....) leading to this conclusion should be adequately explained.

2. Mice

2.1 The authors used CRISPR/Cas9 technology to knock-in the S309W mutation in mice. They report 3 homozygous knock-in mutants and one heterozygous knock-in mouse obtained after IVF from a mosaic founder. Can the authors be sure that the lethality is specific to the knock-in allele and not due for example to off-target effects?.

2.2 Apart from three knock-in homozygotes, the authors also report that: "additional founders generated with the same set of CRISPR reagents were confirmed carried combinations of the S309W mutation and short deletions near the target base pair, most of which also died neonatally". It would be helpful if this can be explained in more detail by indicating in supplemental data how many of these mice were obtained, their genotypes (results from TA-cloning and sequencing) and how many died neonatally or survived the neonatal period. This could provide further evidence of the association of the S309W allele with neonatal lethality. Did the founders with combined genotypes that survived the neonatal period have any phenotype?. It is unfortunate that only one heterozygous S309W mouse was obtained and analyzed. Can additional heterozygotes be obtained from other founders? The analysis of one or two more S309W heterozygotes will reinforce the results on heterozygous mice.

2.3 The characterization of the bone phenotype in mice was conducted by micro-computed tomography. This showed bone alterations in line with the patient phenotype. However, it would be quite informative if in addition the authors can re-use these mice and present histological data of mutant bones.

3. Molecular analysis

3.1 Cell assays and qRT-PCR. The effect of the S309W mutation on alkaline phosphatase staining and expression is surprising because this marker is considerably increased in the two S309W patients. Can the authors comment on this?. It would be more appropriate to refer to chondrocyte maturation assay rather than "osteoblastic differentiation of chondrocytes" (line 338, Material and Methods). It would also be interesting to conduct expression studies in a preosteoblast cell line such as MC3T3-E1.

The identification number of the Taqman assays used in qRT-PCR should be included in material and methods. Considering that cells are retrotransduced with human SP7-cDNA, it would be good to inform that the Sp7 Taqman assay used in these experiments is specific for the endogenous transcript and does not cross-react with human SP7 cDNA. The effect of the S309W mutation in the Sp7 and Col1a1 transcripts is interesting. It should be checked if S309W also leads to an increase in the amount of the corresponding endogenous Sp7 protein and type -I collagen using antibodies.

3.2. EMSA experiments need additional controls to make sure that the band shifts observed in gels

(which in general are weak) truly correspond to the DNA/complexes indicated by the authors. It is not clear if the first lanes of EMSAs (the lanes indicated as negative for all constructs and competitor probe, Figure 4) are assays that include nuclear extracts of untransfected cells. This control is required, or controls using nuclear extracts from cells transfected with the empty expression vector or a non-relevant construct. Antibody supershift assays are other interesting controls useful to validate the identity of the complexes. If the nucleotide probes used in EMSA were obtained from previous publications, add references in material and methods. Also indicate in methods (not only in figure legend) that HEK293 cells were used to obtain the nuclear extracts used in EMSA.

4. Figures.

4.1 In general to avoid confusion, the letter of each panel should be indicated before the legend of the corresponding panel. The number of replicates (n) of each experiment also needs to be indicated in each figure legend.

4.2 Figure 1 L and M. Please indicate if color lines correspond to the patient.

4.3 Figure legend 2. Revise the panel letters in the first sentence of this figure, "wild-type (A,D, G,H)" and in line 431..... "(compare H,K,L)".

4.4 Figure 2M and N. Please expand figure legend of these panels to make it easier for the reader. What is shown in M: tibias, femurs or both?. Indicate if the grey box in N is data from femurs of WT mice of similar age. Figure 2N: The femur mean BMD of the homozygous knock-in mice is lower than normal whereas the single heterozygous mutant has higher mean BMD. Any explanation?

4.5 - Figure 4. Legends of panels G and H mention DLX3. The images show DLX5.

5. Others

5.1. The abstract should not contain citations.

5.2 Material and Methods should include a specific section for genetic analysis including exome sequencing.

5.3 Line 165. There are now three families reported with recessive SP7 mutations and osteogenesis imperfecta. The last one was described by Hayat A et al. Biallelic variants in four genes underlying recessive osteogenesis imperfecta. *Eur J Med Genet.* 2020 Aug;63(8):103954. Please include also this reference.

5.4 Since there are many abbreviations for bone parameters, especially in figures, the authors should add a list of all the abbreviations used in the manuscript or refer to a nomenclature paper (i.e.: David W Dempster DW et al. Standardized Nomenclature, Symbols, and Units for Bone Histomorphometry: A 2012 Update of the Report of the ASBMR Histomorphometry Nomenclature Committee. *J Bone Miner Res.* 2013 January ; 28(1): 2-17.

Reviewer #1 (Remarks to the Author):

Reviewer overall comment: Julian Lui et al. present an interesting manuscript on the role of a novel mutation in SP7/Osterix gene on expression of bone related genes and bone formation in vivo and in vitro. The authors present a set of results implicating a role of mutant SP7/Osx on skeletal development and osteogenesis in humans and mouse. The manuscript tries to provide insightful results including a possible pathological involvement of a sequence specificity modification of zinc-finger motifs. While this concept would draw attention from several different communities, a few issues need further consideration to publish in nature communications.

Reviewer major comment 1: It is not clear whether an Austrian boy is homozygous, heterozygous or mosaic for the mutation. Presentation of genomic sequence data will be helpful for readers to evaluate phenotypes of the patient and mutant mice.

Response: The Austrian boy is heterozygous for the mutation. To clarify this point, as the reviewer suggested, we have added images showing the sequencing results for the proband and parents – both exome sequencing (Supplementary Figure S4A) and Sanger sequencing results (Supplementary Figure S4B). Also, as suggested by the reviewer, we have added images showing the sequencing results for the heterozygous and homozygous mice (Supplementary Figure S6).

Reviewer major comment 2: Heterozygous *Osx* mutants were reported to be normal in general features. Formation of the rib cages in the S309W hets seems to be unaffected in microCT images in fig.2, although they showed delayed bone formation. Therefore, a cause of very low survival rates and very low recovery rates of pups of mutant mice is unclear. The authors should describe the reasons. In addition, this reviewer assumes that this very low recovery rates of mutant pups gave the authors difficulties to analyze phenotypes. As shown in ext. fig.5, production of mutant mice with high mosaicism should be taken into account for further detailed examination including histological studies in postnatal stages. This would also open the door to perform bone histomorphometry in mutant mice.

Response: The reviewer asks us first to discuss the likely reasons for the mortality of mice carrying the mutation. We therefore added the following information to the results section:

“The perinatal mortality observed in the mice carrying either the heterozygous or homozygous S309W SP7 variant may have been due to respiratory failure secondary to the abnormal rib cage with decreased rib length but increased thickness and density, as observed by micro-computed tomography. This mortality limited the number of mice available for study, despite extensive efforts to derive additional mice with the mutation.”

The reviewer also suggests production of mutant mice with high mosaicism to allow further phenotypic characterization including histological studies in postnatal stages. We agree with the reviewer that this approach is of value, with the caveat that the results are more complicated to interpret because the mosaic composition would be variable between different mice and even between different bones in the same mouse. We have only been able to generate two mosaic mice that survived the perinatal period. One was studied at 20 weeks of age and the other at 5 weeks of age. The findings from the mosaic mice are included in the revised manuscript, including the microCT images which are now shown in Supplementary Fig. 7. The phenotype of these mosaic mice was consistent with that of the non-mosaic

newborn mice including short, thickened ribs and thickened clavicle (for the 20-week old mouse; the 5-week old mouse was partially cannibalized and the rib cage was not available for analysis), loss of the normal cortical-trabecular organization, and loss of medullary cavity in long bones.

Reviewer major comment 3: Since generation of mutant lines and a clear description of their phenotypes are requisite to prove an S309W mutation in *Osx* is a cause of the disorder, a thorough examination of the mutant mice is a core of this manuscript. However, only microCT images and bone mineral density distribution are presented. Histological examination of bones formed through intramembranous and endochondral ossification is necessary. Especially, histological studies of growth plate cartilage are required for following reasons. *Osx* is expressed in the mature zone of the mouse growth plate. Expression level of *Col1a1* was increased in growth plate cartilage when *Sox9* was deleted in tissue specific manner (Peter Dy et al., *Developmental Cell* 22, 597–609, 2012). Recent lineage tracing experiments showed growth plate chondrocytes differentiated into osteoblasts in mouse early development (Koji Mizuhashi et al., *Nature*, 563, 254-258, 2018). From these findings, it is reasonable to hypothesize *Osx* in the growth plate has some roles in cartilage to promote endochondral ossification, and hence mutation could affect it. Indeed, the microCT image of the S309W mutant long bone in fig.2K showed delayed ossification (a distance between primary ossification centers are short) in the long bone. Are there any morphological irregularities in growth plate structures? In addition, did mutants show any dysregulation of gene expression patterns of chondrocyte specific genes and aberrant activation of mutant *Osx* and *Col1a1* expression in growth plates? Histological examination and in situ hybridization studies on mutant mice growth plates and primary ossification centers are necessary.

Response:

We think that the reviewer makes an important point about the possible roles of *Osx* in the growth plate, and we have explored the question as the reviewer suggested. We examined the cartilaginous growth plates of wild-type, heterozygous S309W, and homozygous S309W mice histologically using Masson's trichrome stain for the growth plate cartilage and Gomori's trichrome stain for the spongiosa, *Col10a1* in situ hybridization, *Col1a1* RNAscope, and Sp7 RNAscope and immunohistochemistry.

For the growth plate cartilage, the results are shown in Supplementary Fig.S8 and described in the results section as follows:

“We next explored the effects of the S309W mutation on the growth plate by comparing the proximal tibial growth plates of wild-type, heterozygous S309W, and homozygous S309W mice histologically (Supplemental Figure 8). The histological structure of the cartilaginous growth plates looked similar using Masson's trichrome stain. In situ hybridization suggested a normal distribution of *Col10a1*, a marker for hypertrophic differentiation. *Col1a1* mRNA expression by RNAscope and Sp7 protein expression by immunohistochemistry were detected in the perichondrium and primary spongiosa, but not in growth plate chondrocytes, with similar patterns observed in wild-type mice and those carrying the mutation. Prior studies have detected low-level expression of *Col1a1* and Sp7 in growth plate chondrocytes⁶⁻⁸.”

For the spongiosa, immediately below the growth plate, the results are shown in Fig. 2 and described in the results section as follows:

“In contrast, the spongiosa, which is the site of endochondral ossification adjacent to the growth plate, showed marked differences between wild-type mice and those carrying the mutation. In the proximal tibial metaphysis, the heterozygous and homozygous mice showed markedly increased extracellular matrix production, by Masson’s trichrome staining (Figure 2O-O”) but decreased mineralized bone by Gomori’s trichrome staining (Figure 2P,P’), suggesting a failure of the mutant osteoblasts to mineralize bone matrix, a finding similar to that seen in the subject’s bone biopsy. In the mutant mice, Col1a1 mRNA expression occurred in a disorganized pattern with small groups of expressing cells, as opposed the normal sheets of osteoblasts lining the regions of cartilage matrix that are being remodeled into bone matrix (Figure 2Q-Q”). The mutant mice also lacked the normal sheets of osteoblasts that normally contribute to formation of the cortical bone⁹. Similarly, osteocalcin mRNA expression in the cortex, adjacent to the periosteum¹⁰, was greatly diminished in the mutant mice (Fig. 2T-T”). Unexpectedly, the mutant mice appeared to have increased mRNA expression of Sp7 (Fig. 2R-R”).”

Reviewer major comment 4: The authors tried to explain the possible functional alteration of mutant Osx by introducing the recent idea of possible association of Osx with Dlx5. This reviewer is neutral to this concept, however, data presented in fig. 4 is not sufficient enough to reach solid conclusion. First, fig.4B showed relative affinity/binding activities of Sp1 and Osx. The authors used exogenous Sp1 and Osx proteins to show band shifts. These results are perplexing. The authors transfected recombinant genes by retrovirus into HEK cells and preparing nuclear extracts for EMSA. If I did not misunderstand, their nuclear extract is rich in endogenous Sp proteins which could form strong signals in the EMSA. However, a shifted band is only observed in the mixture with SP1-myc nuclear extracts.

Response: We appreciate the reviewer’s thoughtful analysis of the data. We think the likely explanation is that the amount of Sp1 and Osx protein expressed by the retroviral constructs far exceeds endogenous expression, and that EMSA is not sufficiently sensitive to detect a band shift with the relatively low levels of endogenous proteins. This finding is consistent with prior studies (See Hironori Hojo et al., *Developmental Cell* 2016, PMID: 27134141, Figure 6F).

Reviewer major comment 4 (continued): In addition, the mixture with SP1-myc nuclear extracts showed no free probe left in the bottom of the lane, even though Sp7-flag lane showing sifted signals left descent amounts of free probes.

Response: Again, we appreciate the reviewer’s careful analysis of the findings. We repeated the experiment an additional time and again found the same result. This finding is consistent with prior studies (Hojo, *Dev Cell*, 2016, PMID: 27134141)]. We think the likely explanation is that Sp1 binds the GC-box-containing oligonucleotide with greater affinity than does Sp7, and therefore Sp1 leaves less free oligo.

Reviewer major comment 4 (continued): Identities of shifted complexes are not clear. Second, fig4G and H are critical for the authors to describe “in vivo evidence in both mouse and humans that the AT-motif binding specificity of SP7 is physiologically important” in the line 222 and 223. However, again, identities of sifted complexes are not clear (competition with cold probes is sufficient enough).

Response: To address this valid concern, we repeated the EMSA and added antibodies to try to induce supershifting of the bands. The results are shown in Fig. 4B and 4G and described in the results section. The putative Sp1-Myc/ GC-box-oligo band was supershifted by anti-myc. The putative Sp7-Flag/GC-box-oligo band was supershifted by anti-Flag. The putative Dlx5-Myc/Sp7-Flag/ AT-rich-oligo band was

supershifted by both anti-Myc and anti-Flag. We appreciate the reviewer's suggestion which has strengthened the identification of the EMSA bands.

Reviewer major comment 4 (continued): Third, IP experiments showed association of Osx and Dlx5 in the absence of DNA fragments and mutants showed relatively strong binding to Osx, suggesting association of wt/mut Osx and Dlx5 is DNA-binding independent. However, In EMSA, mutant proteins had less activity for complex formation. It is difficult to reconcile these controversial results.

Response: We agree with the reviewer's interpretation of the data that mutant Osx binds well to Dlx5 in the absence of DNA (assessed by co-immunoprecipitation), but mutant Osx, Dlx5, and an AT-rich oligo do not form a complex well, compared to wild-type Osx (assessed by EMSA). We think the likely explanation for these findings is that the mutation in Osx does not impair binding to Osx proteins but it does impair the ability of the Osx-Dlx complex to bind the AT-rich motif. We realize that we did not discuss this issue clearly in the manuscript, and we have therefore added an explanation to the manuscript.

"Our findings therefore suggest that SP7 mutant did not impair the ability of SP7 to interact with DLX proteins (Figure 4F) but rather the ability of the resulting SP7-DLX complex to bind to AT-rich motifs (Figure 4G-H)."

Reviewer major comment 5: ChIP-seq analyses of occupancies of recombinant proteins showed mutant proteins are enriched in the genomic regions where wild type Osx occupancies are low, suggesting that mutation provides changes sequence preferences. These results were obtained from transfected cells in which Dlx5 are not highly expressed. Is it possible to show that mutant proteins are not enriched in the genomic regions where Dlx5 occupancies are high in Dlx5 expressing cells?

Response: We appreciate the suggestion to repeat the ChIP-seq analysis in cells that express sufficient Dlx5 proteins. We agree that the results would be of value. Unfortunately, there are some practical difficulties getting this done within a reasonable time frame. The ChIP-seq data that is already included in the manuscript represents a collaborative effort between the NIH lab and the University of Tokyo lab, in which the NIH investigators performed the retroviral-based expression of mutant or wild-type Sp7, isolated chromatin, and optimized chromatin fragmentation. The fragmented chromatin was then shipped to Tokyo for the immunoprecipitation and RNA-sequencing. The shipping proved problematic because custom delays led to exhaustion of the dry ice, ruining the samples. Because of these problems, the initial ChIP-seq experiment took approximately one year to complete. We anticipate that the shipping problems will be worsened by the Covid-19 pandemic. We have considered consolidating all the procedures in one lab, but ChIP-seq is a difficult approach that requires considerable optimizing. We therefore are concerned that the additional experiment would require many months of additional work. We hope the reviewer and editor will agree with us that, even without this additional experiment, the study is still of value in that it rigorously demonstrates that the observed mutation causes the skeletal disease and provides important insights into the pathophysiology and into the role of Sp7 in osteoblast biology.

Reviewer major comment 6: The authors proposed that S309W mutation caused a high turnover skeletal disorder. However, mechanisms were not described in this manuscript. Even in the in vitro exp in fig.3, effects of mutant proteins on RANKL mRNA expression, Osx protein expression, and type I

collagen expression were not analyzed. The authors should present and discuss the possible mechanisms of a high turnover skeletal phenotype.

Response:

The reviewer makes a good point about the high-turnover state. To explore the possible underlying mechanisms, we expressed either wild-type or mutant Sp7 in growth plate chondrocytes and found that the mutant caused a dramatic increase in Rankl expression (Fig. 3A). This finding suggests that the mutation may similarly increase Rankl expression in vivo, which may stimulate osteoclast differentiation and consequently stimulate bone resorption. We have added this possible mechanism to the discussion, as suggested:

“This finding suggests that the mutation may similarly increase Rankl expression in vivo, which may stimulate osteoclast differentiation and consequently contribute to the increased bone resorption observed in the patient.”

We had previously shown that the mutation increases Col1a1 mRNA expression. As requested by the reviewer, we examined Col1a1 expression at the protein level and found that it was also increased by the Sp7 mutation in both growth plate chondrocytes and mesenchymal stem cells (Fig. 3A).

We had previously shown that the mutant Sp7 increases expression of endogenous, wild-type Sp7 at the mRNA level (Figure 2R-R”). Unfortunately, we were unable to assess this endogenous expression at the protein level because we were unable to find good species-specific antibodies against Osx and therefore an antibody would likely not distinguish between the endogenous (mouse) Sp7 of interest and the exogenous, overexpressed (mutant or wild-type) human Sp7.

Reviewer major comment 7: This reviewer noticed that the patient shows normal linear growth. Nevertheless, as described in comments above, mutant phenotypes including human symptoms could be partially explained by “changes of sequence preferences to GC-rich genomic regions affect chondrocytes differentiation in growth plate”. The discussion in the line 222 and 223 is a bit aggressive when it comes to mechanistic implication of mutant Osx during development.

Response:

We have toned down that conclusion from “Overall, the present study is the first to provide in vivo evidence in both mouse and humans that the AT-motif binding specificity of SP7 is physiologically important” to “Taken together, our findings suggest that the AT-motif binding specificity of SP7 is physiologically important.”

Minor comments

Reviewer minor comment 1: The features of a complex skeletal disorders of the patient are documented to a certain extent. It would be helpful for readers to summarize differences between the Osx-associated high turnover disorder and other form of high turnover disorders in a table.

Response: We thank the reviewer for the suggestion and have added Supplementary Table 2.

Reviewer #2 (Remarks to the Author):

Reviewer overall comment: The goal of this study was to identify and characterize the mutation responsible for a skeletal disorder in a young boy. The authors identified a missense mutation in the Sp7 transcription factor and found that it caused altered DNA binding and gene activation. Overall, the studies are comprehensive and convincing.

Reviewer comment 1: The molecular basis for the increase in bone resorption in the human patient is unclear. Because the mutant mice die so early, it is unlikely that they could be analyzed for such mechanisms. However, expression of the mutant proteins in cells that support osteoclast formation might provide some insight. Alternatively, the authors may possess some chromatin immunoprecipitation results that might shed light on this issue.

Response: The reviewer asks an important question about the mechanisms responsible for increased bone resorption. To explore the possible underlying mechanisms, we expressed either wild-type or mutant Sp7 in vitro in growth plate chondrocytes and MSCs and found that the mutant caused a dramatic increase in Rankl expression at the mRNA level (Fig. 3C) as well as at the protein level in chondrocytes (Fig.3A). Unfortunately, the expression levels in MSCs were much lower than in chondrocytes and we were unable to confidently detect it by western blot. Our findings suggest that the mutation may similarly increase Rankl expression in vivo in osteoblasts, which may stimulate osteoclast differentiation and consequently stimulate bone resorption. We have included these new data in the manuscript and added this possible mechanism to the discussion:

“This finding suggests that the mutation may similarly increase Rankl expression in vivo, which may stimulate osteoclast differentiation and consequently contribute to the increased bone resorption observed in the patient.”

Reviewer comment 2: The legend for figure 2 lacks sufficient detail. A-L appear to be microCT images, but this needs to be stated. Also, it is unclear what the color-coding represents (mineralization?). In panel M it is not clear which bone was analyzed and in panel N it is unclear what genotype is represented by the box and whiskers (wild-type, sample number?)

Response: We appreciate the reviewer pointing out the problems with this figure legend. We have corrected the deficiencies.

Reviewer comment 3: In Extended Data element 2, it is unclear what the red boxes indicate.

Response: Thank you, we have added the missing explanation.

Reviewer #3 (Remarks to the Author):

Reviewer overall comment: The manuscript by Lui JC et al describes a de novo missense mutation (p.S309W) in the transcription factor SP7, in a patient with a bone phenotype characterized by

craniosynostosis, osteosclerosis of the clavicles and spine and bone fragility in the lower extremities. The same mutation has been previously reported as the cause of the disease in another patient with similar phenotype (Whyte MP et al. Bone 2020), thus supporting causality. The authors knocked-in the mutation in mice by CRISPR/Cas9 and observed a neonatal lethal phenotype for both homozygous and heterozygous S309W mutants. The knock-in mice are reported with skeletal features that remind the human phenotype. The authors also performed molecular studies in cultured cells that associate the mutation with a neomorphic function of SP7 due to increased GC-box binding and decreased binding to AT-rich motifs. This work validates the pathogenicity of the S309W mutation, and interestingly, sheds light on the underlying pathomechanism. However, there are parts of the manuscript that need to be more thoroughly explained and experimental procedures that should be revised to strengthen the results.

Reviewer comment 1.1: The patient of this manuscript and the patient reported by Whyte MP et al., both have a marked elevation of ALP. In addition, the patient described by Whyte MP et al. also had increased urinary hydroxyproline but normal deoxypyridinoline. The authors could also check these two parameters in urine to help drawing the biochemical picture of this condition. Classical DXA analysis is a helpful parameter for characterization of bone disease and should be added to the patient's clinical description.

Response: We appreciate the suggestions and have added additional information regarding DXA to the results section:

“The height-adjusted lumbar spine bone mineral density by DXA scan was distinctly increased (+5.8 SDS).”

We did not include the total body less head DXA result because the presence of intramedullary rods would interfere.

We also added the following information regarding bone turnover markers:

“Urine deoxypyridinoline was in the age-specific normal range in two samples (10.2, 12.7 nmol/mmol creatinine, normal range 4.3 – 24.7) and urine hydroxyproline was below the normal range (0.0, 0.1 nmol/mmol creatinine, normal range 0 – 13.0). However, both of these markers of bone resorption and also the DXA measurements may have been affected by the long-term treatment with bisphosphonates which diminish bone resorption.”

Reviewer comment 1.2: The authors indicate that “bone histomorphometry showed increased osteoblast and osteoclast numbers (Figure 1G-J)”. The type of studies and data (whether it was cell counting/bone surface or else....) leading to this conclusion should be adequately explained.

Response: We appreciate the suggestion and have now included a more detailed explanation of the bone histomorphometry findings (see underlined):

“Consistently, bone histomorphometry showed a marked increase in all indices of bone formation (osteoblast bone surface per total bone surface, osteoblast and osteoid surface per bone surface, and osteoid thickness) and of bone resorption (osteoclast surface per bone surface and eroded surface per bone surface) in trabecular bone of the affected patient compared to reference values (Figure 1G-K,

Supplemental Figure 1,2). These results are typical for elevated bone turnover. High bone turnover often results in decreased bone matrix mineralization because newly formed bone packets are removed before they archive full mineralization^{1,2}. Consistent with this expectation, quantitative backscattered electron imaging (qBEI) of trabecular bone showed markedly decreased matrix mineralization with increased heterogeneity (Figure 1L-N).”

Reviewer comment 2.1: The authors used CRISPR/Cas9 technology to knock-in the S309W mutation in mice. They report 3 homozygous knock-in mutants and one heterozygous knock-in mouse obtained after IVF from a mosaic founder. Can the authors be sure that the lethality is specific to the knock-in allele and not due for example to off-target effects?

Response: We agree that this is an important issue to consider. To minimize off-target effects, we used the following approaches:

1. gRNA selection was based on minimal off-target hits with a well-recognized online gRNA selection tool at CRISPRScan.org (PMID:26322839).
2. To avoid overproducing the gRNA, in vitro transcribed gRNA was incubated with SpCas9 protein to form gRNA- Cas9 ribonuclear partials (RNP, PMID: 28410976) which were injected into mouse pronuclei.
3. Injection dosage of the gRNA was titrated carefully to the minimal effective concentration, 1ng/ul in this case, in a gRNA screening assay involving electroporation of a range of dosages (0.1 to 50ng/u) of the gRNA.

With such current state-of-the-art approaches, the off-target effects are reportedly minimal (PMID: 30518723).

Additionally, the fact that a similar phenotype was observed in multiple independent knock-in mice argues against an off-target effect because it would be unlikely for the same off-target change to occur in multiple mice.

We have added this information (underlined) to the methods section:

“Target sequences for guide RNAs (gRNAs) near the target codon with minimal off-target hits were initially selected using the ranking tool CRISPR Design (<http://crispr.mit.edu>) and CRISPRScan.org³⁵, synthesized with the MegaScript T7 kit (Ambion) and efficacy confirmed with Surveyor assay. A single-stranded DNA oligonucleotide carrying the SP7 S309W transversion was designed, based upon the position and orientation of the guide RNA, and synthesized (Integrated DNA Technologies). The combination of the guide RNA with the mutagenic oligonucleotide was further tested for its efficiency, introducing the desired point mutation into the mouse genome by digital-droplet PCR with a pair of probes for wild-type and mutated alleles. The final guide RNA selected (GCGGCAAGGTGTACGGCA, 10ng/μl) was mixed with SpCas9 protein (50 ng/μL; PNA BIO) to form gRNA- Cas9 ribonuclear partials³⁶, and the mutagenic oligo (127mer, GTGAACCTCTTGCCGCAAGTGGCTTTCAGATGCCAAGCTTTCCGTACACCTTGCCGACCCAGGAAAGCCTCTCGCCAGTGTGCCAGCGCAAGTGGCTTTCAGATGCCAAGCTTTCCGTACACCTTGCCGACCCAGGGATGTGGCAGCTGT, 100 ng/μL). The mixture was microinjected into fertilized eggs isolated from the C57BL6/J donor females (Jackson) to generate the knockin allele. Injection dosage of the gRNA was titrated carefully to the minimal effective concentration, 1ng/ul in this case, in a gRNA screening assay”

involving electroporation of a range of dosages (0.1 to 50ng/u) of the gRNA. With such current state-of-the-art approaches, the off-target effects are reportedly minimal³⁷.

We have added the following information to the results section:

“The observation of a similar phenotype in multiple independent knock-in mice argues against an off-target effect of CRISPR/Cas9 because it would be unlikely for the same off-target change to occur in multiple mice.”

Reviewer comment 2.2: Apart from three knock-in homozygotes, the authors also report that: “additional founders generated with the same set of CRISPR reagents were confirmed carried combinations of the S309W mutation and short deletions near the target base pair, most of which also died neonatally”. It would be helpful if this can be explained in more detail by indicating in supplemental data how many of these mice were obtained, their genotypes (results from TA-cloning and sequencing) and how many died neonatally or survived the neonatal period. This could provide further evidence of the association of the S309W allele with neonatal lethality. Did the founders with combined genotypes that survived the neonatal period have any phenotype?

Response: We have added Supplemental Table 1 with the requested information.

Reviewer comment 2.2 (continued): It is unfortunate that only one heterozygous S309W mouse was obtained and analyzed. Can additional heterozygotes be obtained from other founders? The analysis of one or two more S309W heterozygotes will reinforce the results on heterozygous mice.

Response: We agree with the reviewer about the value of additional mice. For that reason, we performed extensive CRISPR-Cas9 pronuclear injections and screened large numbers of mice. We were able to generate two mosaic mice that survived the perinatal period. One was studied at 20 weeks of age and the other at 5 weeks of age. The microCT images are now shown in Fig. S5. The phenotype of these mosaic mice was consistent with that of the non-mosaic newborn mice including short, thickened ribs and thickened clavicle (for the 20-week old mouse; the 5-week old mouse was partially cannibalized and the rib cage was not available for analysis), loss of the normal cortical-trabecular organization, and loss of the medullary cavity in long bones. We agree with the reviewer that more mice would be better and tried very hard to obtain them. However, we think that the similar phenotype observed in the single germline heterozygous mouse, the two mosaic heterozygous mice, and the three germline homozygous mice provide sufficient evidence that this variant in SP7 causes a complex skeletal dysplasia.

Reviewer comment 2.3: The characterization of the bone phenotype in mice was conducted by micro-computed tomography. This showed bone alterations in line with the patient phenotype. However, it would be quite informative if in addition the authors can re-use these mice and present histological data of mutant bones.

Response: We appreciate the suggestion and have performed histological analyses of the mutant bones.

We first histologically examined the cartilaginous growth plates of wild-type, heterozygous S309W, and homozygous S309W mice using Masson’s trichrome stain, Col10a1 in situ hybridization, Col1a1 RNAscope, and Sp7 RNAscope and immunohistochemistry. The results are shown in Supplementary Fig.S8 and described in the results section as follows:

“We next explored the effects of the S309W mutation on the growth plate by comparing the proximal tibial growth plates of wild-type, heterozygous S309W, and homozygous S309W mice histologically (Supplemental Figure 8). The histological structure of the cartilaginous growth plates looked similar using Masson’s trichrome stain. In situ hybridization suggested a normal distribution of Col10a1, a marker for hypertrophic differentiation. Col1a1 mRNA expression by RNAscope and Sp7 protein expression by immunohistochemistry were detected in the perichondrium and primary spongiosa, but not in growth plate chondrocytes, with similar patterns observed in wild-type mice and those carrying the mutation. Prior studies have detected low-level expression of Col1a1 and Sp7 in growth plate chondrocytes⁶⁻⁸.”

We also examined the effects of the mutation in the metaphyseal bone. The results are shown in Fig. 2 and described in the results section as follows:

“In contrast, the spongiosa, which is the site of endochondral ossification adjacent to the growth plate, showed marked differences between wild-type mice and those carrying the mutation. In the proximal tibial metaphysis, the heterozygous and homozygous mice showed markedly increased extracellular matrix production, by Masson’s trichrome staining (Figure 2O-O”) but decreased mineralized bone by Gomori’s trichrome staining (Figure 2P,P’), suggesting a failure of the mutant osteoblasts to mineralize bone matrix, a finding similar to that seen in the subject’s bone biopsy. In the mutant mice, Col1a1 mRNA expression occurred in a disorganized pattern with small groups of expressing cells, as opposed the normal sheets of osteoblasts lining the regions of cartilage matrix that are being remodeled into bone matrix (Figure 2Q-Q”). The mutant mice also lacked the normal sheets of osteoblasts that normally contribute to formation of the cortical bone⁹. Similarly, osteocalcin mRNA expression in the cortex, adjacent to the periosteum¹⁰, was greatly diminished in the mutant mice (Fig. 2T-T”). Unexpectedly, the mutant mice appeared to have increased mRNA expression of Sp7 (Fig. 2R-R”).”

Reviewer comment 3.1: Cell assays and qRT-PCR. The effect of the S309W mutation on alkaline phosphatase staining and expression is surprising because this marker is considerably increased in the two S309W patients. Can the authors comment on this?

Response: The reviewer raises an interesting point which we had not considered. We do not have a strong explanation for this discrepancy between in vitro alkaline phosphatase expression and in vivo circulating alkaline phosphatase activity, and therefore we can only speculate. We added the following material to the results section to address this issue:

“However, the decrease in alkaline phosphatase mRNA expression and staining in vitro does not match the increase in serum alkaline phosphatase activity observed in the proband. We can only speculate that the increase in number of osteoblasts in vivo compensates for the decreased expression per cell or that the cell culture systems do not replicate this aspect of the in vivo osteoblast biology.”

Reviewer comment 3.1 (continued): It would be more appropriate to refer to chondrocyte maturation assay rather than “osteoblastic differentiation of chondrocytes” (line 338, Material and Methods).

Response: We appreciate the suggestion and have changed the methods section accordingly.

Reviewer comment 3.1 (continued): It would also be interesting to conduct expression studies in a preosteoblast cell line such as MC3T3-E1.

Response: We appreciate the suggestion and agree that, in principle, a preosteoblastic cell line, such as MC3T3-E1, would be a good system to study the effects of Sp7. There are, however, some recently reported, practical concerns that different subclones of MC3T3-E1 “bear little functional resemblance to the others, or to primary calvarial osteoblasts.” (Hwang PW, Horton JA. Variable osteogenic performance of MC3T3-E1 subclones impacts their utility as models of osteoblast biology. *Sci Rep* 9, 8299, 2019). These data made us concerned that the effects of the Sp7 mutation might depend on the subclone and might not replicate osteoblasts in vivo. Admittedly the same concerns apply to other cell types. Because of this general concern, in our study, we used two different primary cell types and also used an in vivo (mouse) model. We hope that this triple approach provides sufficient confidence that the results are not an artifact of the experimental system chosen.

Reviewer comment 3.1 (continued):

The identification number of the Taqman assays used in qRT-PCR should be included in material and methods. Considering that cells are retrotransduced with human SP7-cDNA, it would be good to inform that the Sp7 Taqman assay used in these experiments is specific for the endogenous transcript and does not cross-react with human SP7 cDNA.

Response: We appreciate the suggestions. The requested information is now included in the methods section.

Reviewer comment 3.1 (continued): The effect of the S309W mutation in the Sp7 and Col1a1 transcripts is interesting. It should be checked if S309W also leads to an increase in the amount of the corresponding endogenous Sp7 protein and type –I collagen using antibodies.

Response: As requested by the review, we examined Col1a1 expression at the protein level by western blot and found that it was increased by the Sp7 mutation in both growth plate chondrocytes and mesenchymal stem cells (Fig. 3A).

As the reviewer noted, we had previously shown that the mutant Sp7 increases expression of endogenous, wild-type Sp7 at the mRNA level. Unfortunately, we were unable to assess this endogenous expression at the protein level because we were unable to find good species-specific antibodies against Osx and therefore an antibody would likely not distinguish between the endogenous (mouse) Sp7 of interest and the exogenous, overexpressed (mutant or wild-type, human) Sp7.

Reviewer comment 3.2: EMSA experiments need additional controls to make sure that the band shifts observed in gels (which in general are weak) truly correspond to the DNA/complexes indicated by the authors. It is not clear if the first lanes of EMSAs (the lanes indicated as negative for all constructs and competitor probe, Figure 4) are assays that include nuclear extracts of untransfected cells. This control is required, or controls using nuclear extracts from cells transfected with the empty expression vector or a non-relevant construct.

Response: The reviewer raises a good point. We should have stated in the manuscript that the first lanes do indeed contain nuclear extract, which is derived from cells transfected with empty vector. We have now made this clear in the manuscript.

Reviewer comment 3.2 (continued): Antibody supershift assays are other interesting controls useful to validate the identity of the complexes.

Response: We repeated the EMSA and added antibodies to try to induce supershifting of the bands. The results are shown in Fig. 4B and 4G. The putative Sp1-Myc/ GC-box-oligo band was supershifted by anti-myc. The putative Sp7-Flag/GC-box-oligo band was supershifted by anti-Flag. The putative Dlx5-Myc/Sp7-Flag/ AT-rich-oligo band was supershifted by both anti-Myc and anti-Flag. We appreciate the reviewer's suggestion which has strengthened the identification of the EMSA bands.

Reviewer comment 3.2 (continued): If the nucleotide probes used in EMSA were obtained from previous publications, add references in material and methods. Also indicate in methods (not only in figure legend) that HEK293 cells were used to obtain the nuclear extracts used in EMSA.

Response: We appreciate the suggestions. We have now cited the references for the nucleotide probes and indicated in the methods section that HEK293 cells were used for the EMSA studies.

Reviewer comment 4.1: In general to avoid confusion, the letter of each panel should be indicated before the legend of the corresponding panel. The number of replicates (n) of each experiment also needs to be indicated in each figure legend.

Response: Thank you for the suggestion. We made those changes accordingly.

Reviewer comment 4.2: Figure 1 L and M. Please indicate if color lines correspond to the patient.

Response: We have now indicated in the figure legend that the red and green curves represent data from the proband.

Reviewer comment 4.3: Figure legend 2. Revise the panel letters in the first sentence of this figure, "wild-type (A,D, G,H)" and in line 431..... "(compare H,K,L)".

Response: We thank the reviewer for the careful reading and have corrected the legend.

Reviewer comment 4.4: Figure 2M and N. Please expand figure legend of these panels to make it easier for the reader. What is shown in M: tibias, femurs or both?. Indicate if the grey box in N is data from femurs of WT mice of similar age. Figure 2N: The femur mean BMD of the homozygous knock-in mice is lower than normal whereas the single heterozygous mutant has higher mean BMD. Any explanation?

Response: We very much appreciate the suggested improvements and have changed the figure legend accordingly. The reviewer asked for an explanation for why the heterozygous mutant has a higher femoral BMD than wild-type but the homozygous mouse has a lower BMD. Unfortunately, we do not have a good explanation except a speculation that, in the heterozygous mouse, the mutant Sp7 causes upregulation of the wild-type Sp7 allele and that the resultant increase in wild-type Sp7 protein somehow leads to the increase in BMD. Because this explanation is quite speculative, we did not include it in the manuscript, but only noted the difference in BMDs. However, if the reviewer thinks it worthwhile, we could add this potential explanation to the manuscript.

Reviewer comment 4.5: - Figure 4. Legends of panels G and H mention DLX3. The images show DLX5.

Response: We again thank the reviewer for the careful reading and for catching this error. We have corrected it.

Reviewer comment 5.1: The abstract should not contain citations.

Response: Thank you. We have removed citations from the abstract.

Reviewer comment 5.2: Material and Methods should include a specific section for genetic analysis including exome sequencing.

Response: We have added a specific section as suggested:

“Genetic Analysis

The SNP microarray analysis was performed using the Infinium™ OmniExpressExome from Illumina that covers approximately 950,000 SNPs. Exome sequencing was performed at the National Institutes of Health Sequencing Center. A whole-genome library with ~325 base inserts was prepared for each sample using the Kapa DNA Library Preparation Kit (high throughput, with bead) (Kapa Biosystems, Wilmington, MA). Exome capture was performed with the SeqCap EZ Human Exome+UTR Kit v3.0 (Roche Nimblegen, Madison, WI). Each captured exome pool was sequenced on a HiSeq2500 using version 4 chemistry. Data were processed using RTA ver. 1.18.64 and CASAVA 1.8.2. Reads were mapped to National Center for Biotechnology Information (NCBI) build 37 (hg19). All subjects sequencing data were processed in parallel for alignment and variant calling. The combined data were formatted in a single .vs file and analyzed using VarSifter (24) which enables a search for variants in a specific gene and analysis of genotypes using Boolean logic.”

Reviewer comment 5.3: Line 165. There are now three families reported with recessive SP7 mutations and osteogenesis imperfecta. The last one was described by Hayat A et al. Biallelic variants in four genes underlying recessive osteogenesis imperfecta. Eur J Med Genet. 2020 Aug;63(8):103954. Please include also this reference.

Response: We appreciate the reviewer pointing out this new case report and have added a citation.

Reviewer comment 5.4: Since there are many abbreviations for bone parameters, especially in figures, the authors should add a list of all the abbreviations used in the manuscript or refer to a nomenclature paper (i.e.: David W Dempster DW et al. Standardized Nomenclature, Symbols, and Units for Bone Histomorphometry: A 2012 Update of the Report of the ASBMR Histomorphometry Nomenclature Committee. J Bone Miner Res. 2013 January ; 28(1): 2–17.

Response: We have added a citation to the JBMR nomenclature paper.

Reviewers' Comments:

Reviewer #1:

Remarks to the Author:

Comments to NCOMMS-20-19158A

A Neomorphic Variant in the Transcription Factor SP7 Alters Sequence Specificity and Causes a High-Turnover Bone Disorder

The authors responded to this reviewer's comments to some extent. They performed histological examination of growth plates and primary ossification centers of hets and homozygous mutants in fig. 2 and in supplemental figs. In addition, EMSA was performed again in order to strengthen the identification of the complexes in the EMSA. These additional data are helpful for readers to realize the mouse phenotypes and possible functions of the mutant protein.

Since the same mutation in human and the studies on skeletal phenotypes have already been reported (Juvenile Paget's Disease from Heterozygous Mutation of SP7 Encoding Osterix. Michael P. Whyte et al. <https://doi.org/10.1016/j.bone.2020.115364>), the elements of this manuscript are generation of a mouse model, its extensive analyses, and proposition of molecular mechanisms and molecular basis of the disease. However, some parts of the manuscript still need to be more thoroughly explained. There are concerns which were described in this reviewer's previous comments and are not properly addressed.

Overall, this reviewer is not enthusiastic to recommend this manuscript to publish in nature communications.

Major comment#1

Since the exact same mutation in human has already been reported (Juvenile Paget's Disease from Heterozygous Mutation of SP7 Encoding Osterix. Michael P. Whyte et al.

<https://doi.org/10.1016/j.bone.2020.115364>), a core of this manuscript is generation of a mouse model with an S309W mutation in *Osx* and its extensive analyses, especially in postnatal stages. Although they found skeletal phenotypes in newborn hets and homozygous mutant clavicles and long bones, it is difficult to conclude that these mutants faithfully recapitulate the human phenotypes with severe scoliosis, thickened calvarium, craniosynostosis, osteosclerosis of the spine, and recurring fractures in the lower extremities. Skeletal images of one 20wk mosaic mouse with S309W mutation in supplementary fig.7 showed some skeletal irregularities. But the authors provided only CT images without providing detailed histological features. Again, production of mutant mice with different mosaic composition must be generated and extensively analyzed postnatal phenotypes to prove an S309W mutation in *Osx* is responsible for skeletal anomalies found in postnatal life of the patient.

Major comment#2

The other message of this manuscript is possible changes of sequence specificity of mutant *Osx*. Indeed, mutant *Osx* showed an increased binding activity to a GC-rich sequence in the EMSA (fig.4B), and increased occupancies on the chondrocyte genome in ChIP-seq experiments (fig.4E). Indeed, mutant *Osx* was enriched in the genomic regions flanking *Col1a1*, *Sp1*, and *Osterix*. It is important to realize that these regions showed enrichment of wild type *Osx* to some extent. ChIP-seq experiment also showed possible changes of target genes and an aberrant gene expression profile (fig4C). These results seem to be reasonable to conclude that the S309W mutation in the first zinc finger motif of *Osx* is critical for binding activity to a GC-rich sequence and for possible changes of binding site specificity.

The authors then tried to introduce an idea that mutation in *Osx* could affect its ability to associate with DLXs. But the authors provided perplexing results. Although they did EMSA with anti-tag antibodies to strengthen the identification of the complexes in the EMSA, it is very hard to reconcile the EMSA results with the results of IP. The authors tried to respond to my previous comments "Third, IP experiments showed association of *Osx* and DLXs in the absence of DNA fragments and mutants showed relatively strong binding to *Osx*, suggesting association of wt/mut *Osx* and DLXs is DNA-binding independent. However, In EMSA, mutant proteins had less activity for complex formation. It is difficult to reconcile these controversial results". But they responded to my comment by simply adding the sentence "Our findings therefore suggest that SP7 mutant did not impair the ability of SP7 to interact with DLX proteins (Figure 4F) but rather the ability of the resulting SP7-DLX complex to bind to AT-rich motifs (Figure 4G-H)." This sentence did not answer well to my concern. It simply describes what this reviewer found. And It did not explain reasons of

these controversial results at al.

Furthermore, the authors did NOT perform additional ChIP-seq analyses of mutant Osx in Dlx HIGHLY EXPRESSING CELLS. This experiment is necessary to prove the author's hypothesis that the mutant Osx would be less active to form a stable complex with Dlx on genomic regions where the complex regulates possible downstream genes. In this revised manuscript, there is no solid evidence that supports the hypothesis. Without clear explanation and additional ChIP experiments in vitro and in vivo, the author's concluding remarks such as "provides the first in vivo evidence that the affinity of SP7 for AT-rich motifs, unique among SP proteins, is critical for normal osteoblast differentiation." must be removed from the abstract and the main part in the manuscript.

Minor comment#1

Gomori's trichrome staining is not sufficient to evaluate mineralization.

Reviewer #2:

Remarks to the Author:

The authors have nicely address the comments of my previous review.

Reviewer #3:

Remarks to the Author:

The authors have adequately addressed all previous comments and the manuscript has been improved.

Minor comments:

- 1) I would suggest to indicate in the "Genetic Analysis" method section that exome sequencing was performed in both parents and the patient as it can be inferred from Supplemental Figure 4A
- 2) The authors could make sure that all figure panels and supplemental figures are mentioned in the text (Fig. 2G,H,I; Supl. Fig S5)
- 3) For the following sentence: "The mutant mice also lacked the normal sheets of osteoblasts that normally contribute to formation of the cortical bone". Can the corresponding figure showing this defect be indicated after this sentence?
4. Typos: PolyPen (PolyPhen); SERPINF1 (SERPINF1)

REVIEWER COMMENTS

Reviewer #1

Major comment#1

Since the exact same mutation in human has already been reported (Juvenile Paget's Disease from Heterozygous Mutation of SP7 Encoding Osterix. Michael P. Whyte et al. <https://doi.org/10.1016/j.bone.2020.115364>), a core of this manuscript is generation of a mouse model with an S309W mutation in Osx and its extensive analyses, especially in postnatal stages. Although they found skeletal phenotypes in newborn hets and homozygous mutant clavicles and long bones, it is difficult to conclude that these mutants faithfully recapitulate the human phenotypes with severe scoliosis, thickened calvarium, craniosynostosis, osteosclerosis of the spine, and recurring fractures in the lower extremities. Skeletal images of one 20wk mosaic mouse with S309W mutation in supplementary fig.7 showed some skeletal irregularities. But the authors provided only CT images without providing detailed histological features. Again, production of mutant mice with different mosaic composition must be generated and extensively analyzed postnatal phenotypes to prove an S309W mutation in Osx is responsible for skeletal anomalies found in postnatal life of the patient.

Response: As the reviewer requested, we have generated additional mosaic mice and characterized the phenotype.

In the earlier drafts of the paper, we characterized mice with a heterozygous germline mutation at birth but could not evaluate the phenotype at later ages because these mice die soon after birth. As the reviewer notes, generating mosaic mice might allow survival to adulthood and thus help characterize the postnatal phenotypic effects of the mutation. Unfortunately, generating these mosaic mice has been extraordinarily difficult. We have made multiple attempts to generate these mice and met with limited success. The reason for this difficulty is that we are generating a single base-pair substitution in the genome using CRISPR-Cas9. This procedure often yields other nearby genomic alterations, particularly indels. Consequently, generating a mosaic mouse with exactly the right genotype is a very rare event. Since receiving the second review, we have tried yet again to generate mosaic mice with the S309W variant.

With these extensive efforts, we have now generated and characterized four mosaic mice that survived into postnatal life (3.5, 5, 15, and 20 weeks of age). As requested by the reviewer, we have used both microCT and histology to characterize the bone phenotype (see Fig. 2U-W and Fig S7). Three of the mice have a complex genotype that includes some combination of wild-type, S309W mutation, and other alleles. However, the phenotype is remarkably consistent, both with each other and with the mice carrying heterozygous germline S309W mutations evaluated at birth. This consistency suggests that we have correctly determined the skeletal phenotype of the mutation.

The mouse phenotype included ribs and clavicles which were decreased in length but increased in thickness and density, which is reminiscent of the clavicle osteosclerosis in the patient. In the long bones of the mouse extremities and ribs, cortical bone formation appeared impaired but there was a marked increase in trabecular bone near the center of the diaphysis and therefore a diminished medullary

cavity. Histologically, the mice showed markedly increased extracellular matrix production but decreased mineralized bone, suggesting a failure of the mutant osteoblasts to mineralize bone matrix, a finding similar to that seen in the human subject's bone biopsy.

The primary purpose of generating the mouse models was to determine whether the S309W missense mutation causes a skeletal disorder and whether the phenotype is qualitatively different from that of loss-of-function mutations, supporting the hypotheses that 1) the mutation is causative in the human and 2) the mutation does not simply cause a loss of Sp7 function. The findings supported both these hypotheses. The exact phenotype of the mouse and human were similar in some ways, different in others. For example, the mouse did not show craniosynostosis. Differences in phenotype between these two species are seen for numerous genetic defects (PMID: 27121451, 11917158, 18199409). Thus, we think that some phenotypic differences between the mouse and human are to be expected and do not invalidate the conclusions.

In the results section, we have described the phenotype as follows:

“We also performed micro-computed tomography and histological analyses of the four mosaic mice that survived postnatally (two of which died at 24 days and 5 weeks of age, two others were sacrificed at 15 weeks and 20 weeks) and found skeletal phenotypes, including thickened clavicle and ribs, uneven cortical thickness, and reduced medullary cavity, similar to the heterozygous and homozygous mutants.”

In the discussion, we address the similarities and difference between the mouse and the human. as follows:

“We created a mouse with the orthologous missense variant in Sp7, which produced a complex skeletal phenotype, confirming that the variant is pathogenic, but dissimilar to the Sp7 knockout mouse, both in inheritance and phenotype, confirming that the variant does not cause a simple loss of function. The human and mouse phenotypes showed some differences. For example, cranial hyperostosis in our patient is not observed in the knock-in mouse. These differences might be due to differences in species and/or the young age of the mice.”

Major comment#2

The other message of this manuscript is possible changes of sequence specificity of mutant Osx. Indeed, mutant Osx showed an increased binding activity to a GC-rich sequence in the EMSA (fig.4B), and increased occupancies on the chondrocyte genome in ChIP-seq experiments (fig.4E). Indeed, mutant Osx was enriched in the genomic regions flanking Col1a1, Sp1, and Osterix. an. It is important to realize that these regions showed enrichment of wild type Osx to some extent. ChIP-seq experiment also showed possible changes of target genes and an aberrant gene expression profile (fig4C). These results seem to be reasonable to conclude that the S309W mutation in the first zinc finger motif of Osx is critical for binding activity to a GC-rich sequence and for possible changes of binding site specificity.

The authors then tried to introduce an idea that mutation in Osx could affect its ability to associate with Dlx5. But the authors provided perplexing results. Although they did EMSA with anti-tag antibodies to strengthen the identification of the complexes in the EMSA, it is very hard to reconcile the EMSA results with the results of IP. The authors tried to respond to my previous comments “Third, IP experiments showed association of Osx and Dlx5 in the absence of DNA fragments and mutants showed relatively strong binding to Osx, suggesting association of wt/mut Osx and Dlx5 is DNA-binding independent.

However, In EMSA, mutant proteins had less activity for complex formation. It is difficult to reconcile these controversial results". But they responded to my comment by simply adding the sentence "Our findings therefore suggest that SP7 mutant did not impair the ability of SP7 to interact with DLX proteins (Figure 4F) but rather the ability of the resulting SP7-DLX complex to bind to AT-rich motifs (Figure 4G-H)." This sentence did not answer well to my concern. It simply describes what this reviewer found. And It did not explain reasons of these controversial results at al. Furthermore, the authors did NOT perform additional ChIP-seq analyses of mutant Osx in Dlx HIGHLY EXPRESSING CELLS. This experiment is necessary to prove the author's hypothesis that the mutant Osx would be less active to form a stable complex with Dlx on genomic regions where the complex regulates possible downstream genes. In this revised manuscript, there is no solid evidence that supports the hypothesis. Without clear explanation and additional ChIP experiments in vitro and in vivo, the author's concluding remarks such as "provides the first in vivo evidence that the affinity of SP7 for AT-rich motifs, unique among SP proteins, is critical for normal osteoblast differentiation." must be removed from the abstract and the main part in the manuscript.

Response: The reviewer pointed out that our Co-IP experiments and EMSA experiments showed results that were difficult to reconcile. Specifically, our Co-IP experiments suggested that both wild-type and mutant SP7 were able to bind DLX proteins, but in our EMSA experiments, we showed that the mutant SP7 did not demonstrate binding to AT-rich DNA motifs. We previous only acknowledged this discrepancy without trying to investigate further.

We have now performed additional experiment to understand this discrepancy between the two experiments.

First, we aimed to determine better the effect of the SP7 mutation on binding to DLX proteins. We first repeated the Co-IP experiment for DLX5 and SP7 proteins and still observed similar binding (Fig 5A). We then reasoned that, while our original Co-IP showed binding of DLX to wild-type or mutant SP7 when done individually, it was not ideal for comparing their binding ability. We therefore performed new co-IP experiments in which we expressed both wild-type and mutant SP7 in the presence of DLX3/5. In this situation, we tagged the wild-type SP7 with a FLAG-tag and the mutant SP7 with an HA-tag (or vice versa), and let them compete for binding to DLX. We found that, regardless of the tag used, the mutation decreased the ability of SP7 to compete for DLX proteins (see Fig. 5B), suggesting that the mutation decreases the affinity of SP7 for DLX3/5. Our previous conclusion that the mutation did not affect DLX protein binding is therefore inaccurate.

We describe the new findings in the Results section as follows:

"To compare the binding ability of mutant and wild-type SP7, we next performed co-IP experiments in which we expressed both wild-type and mutant SP7 in the presence of DLX3/5. In this situation, we tagged the wild-type SP7 with FLAG-tag and the mutant SP7 with an HA-tag (or vice versa) and let them compete for binding to DLX. We found that, regardless of the tag used, the mutation decreased the ability of SP7 to compete for DLX proteins (see Fig. 5B), suggesting that the mutation decreases the affinity of SP7 for DLX3/5."

Secondly, our previous EMSA showed the absence of AT-motif binding by the mutant SP7, which is not consistent with the fact that mutant SP7 does interact with DLX, although with a decreased affinity. To investigate further, we used another approach complimentary to the EMSA, in which we allowed a

biotin-labeled AT-motif oligo to interact with DLX5 and SP7 (either wild-type or mutant), and performed co-IP with streptavidin beads followed by western blot. In this new experiment, we found that the SP7 mutant showed diminished, but not completely absent, binding to the AT-rich oligo in the presence of DLX3/5 (See Fig. 5C). The findings therefore add to the previous results with EMSA and suggest that the mutation did not completely abolish but rather decreased the SP7-DLX-AT-motif interaction. Taken together with the competition co-IP results mentioned above, the combined findings provide us with a more accurate conclusion, that the mutation affects the ability of SP7 to bind DLX proteins and consequently the ability of SP7 to interact, through DLX3/5, with the AT-rich binding motif. We would like to thank the reviewer for suggesting the further investigation, which led to these findings.

We describe the new findings in the Results section as follows:

“We further assessed the ability of wild-type and mutant SP7 to bind the AT-rich motif by oligo-co-immunoprecipitation. A biotin-labeled oligonucleotide containing the AT-rich motif was allowed to interact with DLX5 and SP7 (either wild-type or mutant), and then co-immunoprecipitation was performed with streptavidin beads, followed by western blot. We found that the mutant SP7 showed diminished, but not completely absent, binding to the AT-rich oligo in the presence of DLX3/5 (Fig. 5C). The findings, together with the EMSA results, suggest that the mutation did not completely abolish but rather decreases the SP7-DLX-AT-motif interaction. Taken together with the competition co-IP results mentioned above, the combined findings indicate that the mutation affects the ability of SP7 to bind DLX proteins and consequently the ability of SP7 to interact, through DLX3/5, with the AT-rich binding motif.”

As the review also suggested, we have now performed ChIP-seq in MC3T3 cells transfected with a plasmid expressing DLX5. Our new data show that, in these MC3T3 cells, both wild-type and mutant SP7 were able to bind to chromatin, but wild-type SP7 generated more peaks than the mutant (Fig. 5F), suggesting a decreased binding of mutant SP7 in the presence of DLX proteins. Interestingly, this contrasts with the ChIP-Seq findings in chondrocytes, (which express little DLX), in which mutant SP7 showed increased binding to a large number of genomic regions across the genome that were not bound by wild-type. In the MC3T3 cells, analysis of binding motifs showed that the shared peaks and wild-type specific peaks were very similarly AT-motif enriched sequences (Fig 5G), while mutant-specific peaks were enriched with sequences different from the shared or wild-type peaks, indicating a change in binding specificity in the mutant. This finding also contrasts with the results in chondrocytes where the major binding motifs found were GC-rich sequences. Examples of genomic regions recognized by the wild-type Sp7 but not the mutant in the MC3T3 cells includes a previously identified region in the Notch2 gene (Fig5H, also see Hironori et al, 2016 Dev Cell). Taken together, the ChIP-Seq findings in chondrocytes and MC3T3 cells. confirm that, in the absence of DLX proteins, wild-type SP7 tends to recognize GC-rich sequences, but, in the presence of DLX proteins, wild-type SP7 tends to recognize instead AT-motif sequences. The combined findings also indicate that the mutation in SP7 increases binding to GC-rich regions without DLX proteins, but decreases binding to AT-rich motifs with DLX proteins, supporting our in vitro binding results from EMSA and co-IP experiments. We very much appreciate the reviewer’s suggestions which led to these important new findings.

We describe the new findings in the Results section as follows:

“We next performed ChIP-seq in MC3T3 cells transfected with a DLX5-expressing plasmid. Both wild-type and mutant SP7 were able to bind to chromatin, but wild-type SP7 generated more peaks than the mutant (Figure 5F), suggesting that, in the presence of DLX proteins, the mutation decreases genomic binding of mutant SP7. This finding contrasts with the findings in chondrocytes, (which express little

DLX), in which mutant SP7 showed increased binding to a large number of genomic regions not bound by wild-type SP7. In these MC3T3 cells, analysis of binding motifs showed that the common peaks and wild-type-specific peaks were very similarly AT-motif enriched sequences (Figure 5G), while mutant-specific peaks were enriched with sequences different from the shared or wild-type peaks, indicating the mutation alters binding specificity of SP7. This finding also contrasts with the results in chondrocytes where the major binding motifs found were GC-rich sequences. Examples of genomic regions recognized by the wild-type Sp7 but not the mutant includes a region in the Notch2 gene previously reported to bind SP7 (Figure 5H). Taken together, the CHIP-Seq findings in chondrocytes and MC3T3 cells confirm that, in the absence of DLX proteins, wild-type SP7 tends to recognize GC-rich sequences; but in the presence of DLX proteins, wild-type SP7 tends to recognize instead AT-motif sequences. The combined findings also indicate that, in the absence of DLX proteins, the S309W mutation increases binding to GC-rich regions, but, in the presence of DLX proteins, the mutation decreases binding to AT-rich motifs, supporting our in vitro binding results from the EMSA and co-IP experiments.”

Minor comment#1

Gomori's trichrome staining is not sufficient to evaluate mineralization.

Response: We agree with the reviewer that Gomori's trichrome stain does not allow for a quantitative assessment of bone mineralization per volume. However, as noted by Urlaub KM et al. (PMID: 29883588), “The Gomori trichrome stain identifies bone in turquoise and osteoid and nonmineralized matrix in red, allowing for digital color analysis.” This ability to distinguish mineralized from unmineralized matrix is traditionally considered sufficient for histological analysis where the endpoint is the amount of unmineralized bone in a section, rather than the concentration of calcium per volume. To address this important issue, we have amended the results as follows:

“In the proximal tibial metaphysis, the heterozygous and homozygous mice showed markedly increased extracellular matrix production, by Masson's trichrome staining (Figure 2O-O”). However, much of this matrix was unmineralized as demonstrated by Gomori's trichrome staining (Figure 2P,P'). Gomori staining does not provide a quantitative measure of mineral concentration, but the histological finding of decreased mineralized bone is reinforced by the quantitative analysis by micro-computed tomography described above. Together, the results suggest a failure of the mutant osteoblasts to mineralize bone matrix, a finding similar to that seen in the subject's bone biopsy.”

Reviewer #2 (Remarks to the Author):

The authors have nicely address the comments of my previous review.

Reviewer #3 (Remarks to the Author):

The authors have adequately addressed all previous comments and the manuscript has been improved.

Minor comments:

1) I would suggest to indicate in the "Genetic Analysis" method section that exome sequencing was performed in both parents and the patient as it can be inferred from Supplemental Figure 4A

Response: We have added this information as suggested.

2) The authors could make sure that all figure panels and supplemental figures are mentioned in the text (Fig. 2G,H,I; Supl. Fig S5)

Response: We have corrected the omissions as suggested.

3) For the following sentence: "The mutant mice also lacked the normal sheets of osteoblasts that normally contribute to formation of the cortical bone". Can the corresponding figure showing this defect be indicated after this sentence?

Response: We have added the figure reference as suggested.

4. Typos: PolyPen (PolyPhen); SERPiNF1 (SERPINF1)

Response: We have corrected the typo. We thank the reviewer for the careful reading of the manuscript.

Reviewers' Comments:

Reviewer #1:

Remarks to the Author:

The authors have adequately addressed all previous comments. Additional experiments to show molecular mechanisms of mutant SP7/OSX would be helpful for readers. The manuscript has been improved.

REVIEWER COMMENTS

Reviewer #1

The authors have adequately addressed all previous comments. Additional experiments to show molecular mechanisms of mutant SP7/OSX would be helpful for readers. The manuscript has been improved.

Response:

We thank the reviewer for the positive comment.